# Repeated passive heat treatment increases muscle tissue capillarization, but does not affect postprandial muscle protein synthesis rates in healthy older adults

Cas J. Fuchs[1] , Milan W. Betz[1] , Heather L. Petrick[1], Jil Weber[1] , Joan M. Senden[1], Floris K. Hendriks[1] , Julia L.M. Bels[2] , Luc J.C. van Loon[1] and Tim Snijders[1]

[1]*Department of Human Biology, Research Institute of Nutrition and Translational Research in Metabolism, Maastricht University Medical Centre+, Maastricht, the Netherlands*
[2]*Department of Intensive Care, Research Institute of Nutrition and Translational Research in Metabolism, Maastricht University Medical Centre+, Maastricht, the Netherlands*

Handling Editors: Paul Greenhaff & Christopher Sundberg

The peer review history is available in the Supporting Information section of this article (https://doi.org/10.1113/JP286986#support-information-section).

The Journal of Physiology

**Abstract figure legend** Eight weeks of repeated passive heat treatment (PHT) increases muscle tissue capillarization, but does not improve basal or postprandial muscle microvascular blood flow or muscle protein synthesis rates in healthy, older adults (*n* = 14; 9 males, 5 females). In line with this, merely the application of PHT does not induce muscle (fibre) hypertrophy or increase leg muscle strength. Furthermore, prolonged PHT does not improve mitochondrial function or increase mitochondrial content, but does improve various markers of metabolic and cardiovascular health. Created with BioRender.com.

**Abstract** Prolonged passive heat treatment (PHT) has been suggested to trigger skeletal muscle adaptations that may improve muscle maintenance in older individuals. To assess the effects of PHT on skeletal muscle tissue capillarization, perfusion capacity, protein synthesis rates, hypertrophy and leg strength, 14 older adults (9 males, 5 females; 73 ± 6 years) underwent 8 weeks of PHT (infrared sauna: 3× per week, 45 min at ∼60°C). Before and after PHT we collected muscle biopsies to assess skeletal muscle capillarization and fibre cross-sectional area (CSA). Basal and postprandial muscle tissue perfusion kinetics and protein synthesis rates were assessed using contrast-enhanced ultrasound and primed continuous L-[*ring*-$^{13}C_6$]phenylalanine infusions, respectively. One-repetition maximum (1RM) leg strength and vastus lateralis muscle CSA were assessed. Type I and type II muscle fibre capillarization strongly increased following PHT (capillary-to-fibre perimeter exchange index: +31 ± 18 and +33 ± 30%, respectively; $P < 0.001$). No changes were observed in basal (0.24 ± 0.27 *vs.* 0.18 ± 0.11 AU; $P = 0.266$) or postprandial (0.20 ± 0.12 *vs.* 0.18 ± 0.14 AU; $P = 0.717$) microvascular blood flow following PHT. Basal (0.048 ± 0.014 *vs.* 0.051 ± 0.019%/h; $P = 0.630$) and postprandial (0.041 ± 0.012 *vs.* 0.051 ± 0.024%/h; $P = 0.199$) muscle protein synthesis rates did not change in response to prolonged PHT. Furthermore, no changes in vastus lateralis muscle CSA (15.3 ± 4.6 *vs.* 15.2 ± 4.6 cm$^2$; $P = 0.768$) or 1RM leg strength (46 ± 12 *vs.* 47 ± 12 kg; $P = 0.087$) were observed over time. In conclusion, prolonged PHT increases muscle tissue capillarization but this does not improve muscle microvascular blood flow or increase muscle protein synthesis rates in healthy, older adults. Prolonged PHT does not induce skeletal muscle hypertrophy or increase leg strength in healthy, older adults.

(Received 28 May 2024; accepted after revision 13 September 2024; first published online 7 October 2024)

**Corresponding author** T. Snijders: Department of Human Biology, Faculty of Health, Medicine and Life Sciences, Maastricht University, PO Box 616, 6200 MD Maastricht, the Netherlands. Email: tim.snijders@maastrichtuniversity.nl

## Key points

- Repeated exposure to heat has been suggested to trigger skeletal muscle adaptive responses.
- We investigated the effect of 8 weeks of whole-body passive heat treatment (PHT; infrared sauna: 3× per week for 45 min at ∼60°C) on skeletal muscle tissue capillarization, perfusion capacity, basal, and postprandial muscle protein synthesis rates, muscle (fibre) hypertrophy, and leg strength in healthy, older adults.
- Prolonged PHT increases muscle tissue capillarization, but this does not improve muscle micro-vascular blood flow or increase muscle protein synthesis rates.
- Despite increases in muscle tissue capillarization, prolonged PHT does not suffice to induce skeletal muscle hypertrophy or increase leg strength in healthy, older adults.

## Introduction

Food intake, and dietary protein ingestion in particular, increases muscle protein synthesis rates (Fuchs et al., 2019; Groen et al., 2015; Rennie et al., 2002; Wolfe, 2002) and, as such, forms a key factor in skeletal muscle mass maintenance. The muscle protein synthetic response to food intake is blunted in older individuals (Cuthbertson

**Cas Fuchs** is a post-doctoral researcher and teacher at Maastricht University Medical Centre+ (MUMC+). He obtained his PhD from Maastricht University, where he investigated strategies for post-exercise recovery, with a particular emphasis on carbohydrate and protein intake, as well as cooling and heating interventions. Currently, Dr Fuchs utilizes advanced techniques, such as stable isotope tracer methodology and Magnetic Resonance Imaging (MRI)/Spectroscopy (MRS), to thoroughly study human physiology, with a focus on exercise, nutrition, and body composition. Current research projects include the non-invasive assessment of liver and muscle glycogen and the impact of cooling and heating on muscle physiology.

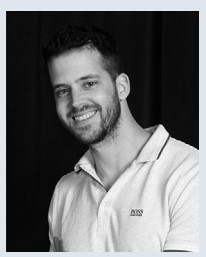

et al., 2005; Katsanos et al., 2005; Wall et al., 2015), and strongly contributes to the loss of skeletal muscle mass observed with ageing (Cuthbertson et al., 2005; Fuchs et al., 2023; Wilkinson et al., 2018). This blunted response to protein feeding is likely multifactorial and includes a compromised postprandial increase in skeletal muscle tissue perfusion (Phillips et al., 2012; Phillips et al., 2015; Timmerman et al., 2010a; Timmerman et al., 2010b).

Adequate muscle tissue perfusion is essential in skeletal muscle mass maintenance and growth, as it allows the rapid postprandial delivery of amino acids, nutrients and growth factors to the muscle fibre, thereby stimulating muscle protein synthesis rates. Arterial blood flow has been reported to be significantly reduced under both fasting and postprandial conditions in older adults (Dinenno et al., 1999; Donato et al., 2006; Phillips et al., 2012; Skilton et al., 2005). This age-related reduction in blood flow has been reported to be independent of muscle mass and may be related to chronic vasoconstriction, lower $O_2$ demands and decreased endothelial wall function (Dinenno et al., 2001; Vincent et al., 2006). The delivery of nutrients to the muscle fibre is ultimately limited by the surface area of the microvascular bed (i.e. capillaries) (Pittman, 1995; Segal, 2005). We (Groen et al., 2014; Nederveen et al., 2016; Verdijk et al., 2016) as well as others (Coggan et al., 1992; Croley et al., 2005) have previously shown that muscle fibre capillarization is reduced in senescent muscle, particularly surrounding the type II muscle fibres. Therefore, increasing muscle fibre capillarization and as such muscle tissue perfusion capacity, may be an effective way to augment postprandial muscle protein synthesis rates in older adults.

Although prolonged (aerobic) exercise training is considered to be the most effective intervention strategy to increase muscle fibre capillarization, and thereby improve tissue perfusion capacity (Gavin et al., 2007; Jensen et al., 2004), its application is not always effective due to poor adherence or compliance to prescribed exercise programmes. Therefore, we should also explore alternative therapeutic intervention strategies that may improve muscle tissue perfusion capacity. Passive heat treatment (PHT), which includes methods such as hot baths, steam rooms, traditional and infrared saunas, may be a promising alternative. Generally, PHT offers a range of health benefits (Patrick & Johnson, 2021), including reduced risks of cardiovascular disease and all-cause mortality (Laukkanen et al., 2015). Additionally, recent studies suggest that PHT may positively impact skeletal muscle adaptations (Kim et al., 2020a), with improvements in skeletal muscle mitochondrial adaptations (Marchant et al., 2023) and increases in muscle mass and/or strength, although the latter remains largely inconclusive (Goto et al., 2011; Kim et al., 2020b; Labidi et al., 2021). In addition, muscle fibre capillarization has been reported to increase following

6 weeks of PHT in healthy young sedentary adults, with the magnitude of change not different compared with equal time spent performing aerobic exercise (Hesketh et al., 2019). However, whether prolonged PHT can induce improvements in skeletal muscle fibre capillarization and perfusion capacity, and subsequently augment basal and/or postprandial muscle protein synthesis rates in older human adults remains to be assessed. Therefore, in the present study, we recruited 14 healthy older adults in whom we investigated the impact of 8 weeks of PHT (3× per week) on muscle fibre capillarization, muscle fibre hypertrophy, and basal and postprandial muscle tissue perfusion kinetics as well as basal and postprandial muscle protein synthesis rates.

## Methods

### Participants

Fourteen healthy older adults (male/female; 65–85 years, body mass index (BMI) >18.5 and <30 kg/m²) with an average age of 73 ± 6 years volunteered to participate in this trial (ratio male/female: 9/5). The trial was registered at ClinicalTrials.gov (NCT05129995) and was conducted at Maastricht University Medical Centre+ in Maastricht, the Netherlands. All participants were informed on the purpose of the study, the experimental procedures and possible risks before providing informed written consent to participate. The procedures that were followed were approved by the Medical Research Ethics Committee Academic Hospital Maastricht/Maastricht University (METC 21−072), and in accordance with the *Declaration of Helsinki* of 1975 as revised in October 2013. The study was independently monitored by the Clinical Trial Centre Maastricht.

### Preliminary screening

Participants underwent an initial screening session to assess eligibility. Participants were deemed healthy based on their responses to a medical questionnaire. Exclusion criteria for participation were having gastro-intestinal, musculoskeletal, metabolic or pulmonary disorders; diagnosis of cardiovascular disease, kidney disease or rheumatoid arthritis; smokers; currently (or within the past 3 months) performing progressive exercise training; had been a frequent (more than once per week) user of infrared or traditional sauna in the past 3 months; had more than 5% body mass change over the past 6 months; were using medication that affected protein metabolism or using gastric acid suppressing medication; participated in recent (<1 year) amino acid tracer studies; donated blood in the past 2 months; had any known allergic reactions to the ultrasound contrast agent; were intolerant

to the investigated food products; or were following a selective diet (e.g. vegan diet). An additional check of the lungs and heart was performed using a stethoscope by an experienced physician. Subsequently, we determined height (m), body mass (kg; Seca 799, Seca GmbH & Co., Hamburg, Germany), blood pressure (mmHg) and heart rate (Omron HEM-907, Omron Healthcare, Kyoto, Japan) in all eligible participants. Next, body composition was assessed by dual-energy X-ray absorptiometry (DXA, Discovery A, Hologic, Marlborough, MA, USA; National Health and Nutrition Examination Survey – body composition analysis (NHANES BCA) enabled) together with a panoramic ultrasound cross-sectional area (CSA) assessment of M. vastus lateralis at one-third distance between the superior patellar border and the anterior superior iliac spine (Affinity 70G; using a linear array probe: eL18-4, Philips, Amsterdam, the Netherlands). Ultrasound images were later analysed by manual tracing using ImageJ software (version 2.0.0; National Institutes of Health, Bethesda, MD,

USA). A Short Physical Performance Battery (SPPB) was performed together with a Timed Up and Go (TUG) test and hand grip strength. Handgrip strength was performed on both the left and right hand and the highest value out of three attempts was recorded. The average of both left and right hand was used for analysis. Furthermore, one-repetition maximum (1RM) leg muscle strength was assessed for both the leg press and extension machines, as described previously (Fuchs, Kouw et al., 2020). Finally, the SQUASH (Short QUestionnaire to ASses Health enhancing physical activity) questionnaire was used to evaluate habitual physical activity level (Wendel-Vos et al., 2003). The screening session and the first experimental trial were separated by at least 5 days.

## Study design

Figure 1 shows a schematic overview of the study design. Following screening and pre-measurements, all

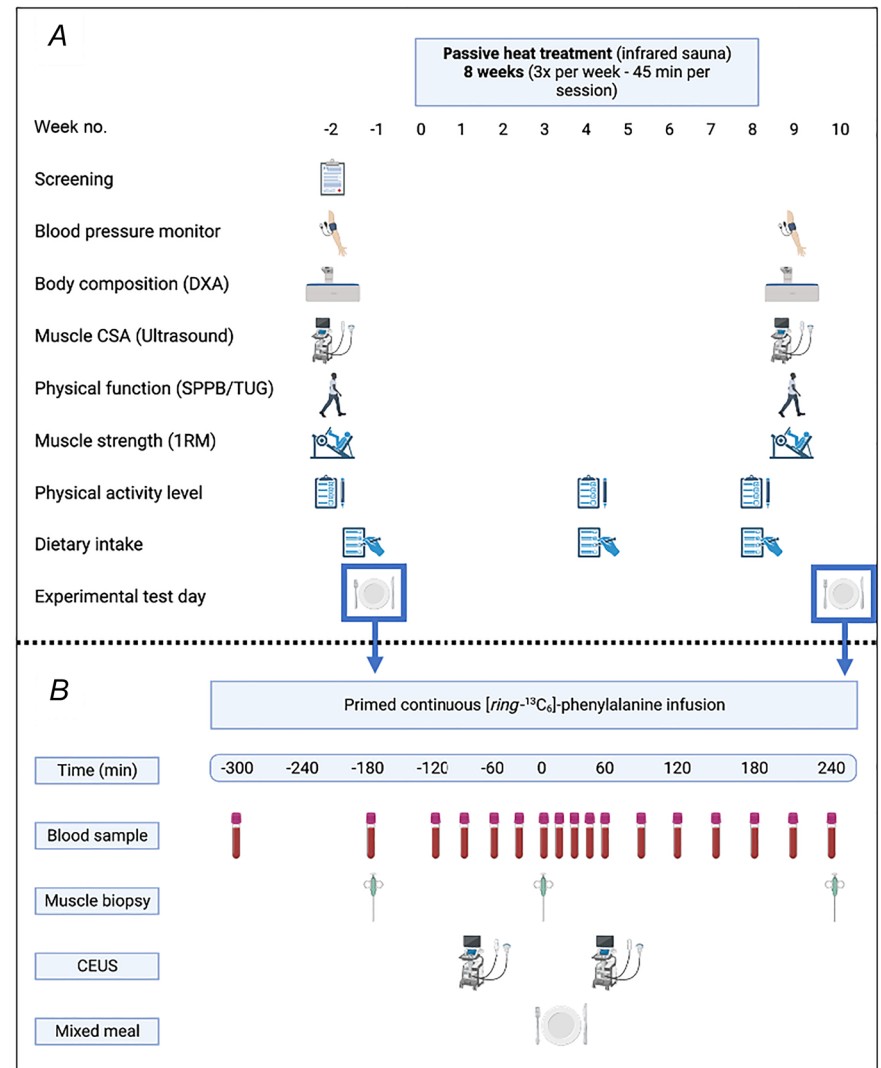

**Figure 1. Experimental protocol of the entire research study (*A*) and the experimental test day (*B*)**
The same experimental test day protocol (*B*) was performed before and after 8 weeks of passive heat treatment. During the experimental test days (*B*), the mixed meal was ingested at *t* = 0 min. CEUS, contrast-enhanced ultrasound; CSA, cross-sectional area; DXA, dual-energy X-ray absorptiometry; SPPB, Short Physical Performance Battery; TUG, Timed Up and Go test. Partly created with BioRender.com. [Colour figure can be viewed at wileyonlinelibrary.com]

participants participated in a single experimental test day (9 h) in which their muscle protein synthetic and perfusion response were assessed before and after ingesting a wholefood protein-rich meal (experimental test day, Fig. 1*B*). Approximately 1 week after the first experimental test day, participants started with the 8-week PHT protocol (3× per week; 45 min per session). Habitual physical activity level (SQUASH questionnaire) and dietary intake record was completed before and in week 4 and 8 of the PHT intervention period. Following the 8-week PHT intervention, all pre-measurements (i.e. blood pressure, heart rate, body composition, M. vastus lateralis CSA, SPPB, TUG, handgrip and 1RM leg muscle strength) and the 9-h experimental test day were repeated.

### Diet and physical activity

Participants refrained from sports, strenuous physical activities and alcohol consumption for 2 days prior to each of the two experimental test days. All participants filled out a food intake diary for two days prior to the first experimental test day as well as in week 4 of the PHT intervention period to assess potential changes in habitual dietary intake over time. For the 2 days prior to the second experimental test day, participants replicated the foods consumed and activities performed in the two days leading up to the first experimental test day. Energy and macronutrient intakes were calculated with the use of the Dutch Nutrients Database (NEVO-online version 2019/6.0; https://nevo-online.rivm.nl/).

The evening before each experimental test day, all participants consumed the same standardized meal containing 2.3 MJ, with 65% energy provided as carbohydrate, 20% as fat, and 15% as protein, before 22.00 h, after which they remained fasted.

### Experimental test day

A schematic representation of the experimental test day is shown in Fig. 1*B*. Participants performed an identical meal test day before and after 8 weeks of PHT. At 07.30 h, participants arrived at the laboratory in an overnight fasted state. A catheter was inserted into an antecubital vein for stable isotope amino acid infusion, while a second catheter was inserted retrogradely into a dorsal hand vein of the contralateral arm for arterialized blood sampling. To obtain arterialized blood samples, the hand was placed in a hotbox (60°C) for 10 min prior to each blood sample collection (Abumrad et al., 1981). After taking a baseline blood sample (t = −300 min), the plasma phenylalanine pool was primed with a single dose of L-[*ring*-$^{13}C_6$]phenylalanine (2.25 μmol/kg). Thereafter, a continuous intravenous infusion of

L-[*ring*-$^{13}C_6$]phenylalanine (0.05 μmol/kg/min) was initiated (t = −300 min) using a calibrated Space$^{plus}$ Infusomat pump (B. Braun, Melsungen, Germany). While resting in a supine position, arterialized blood samples were collected at t = −180, −120, −90, −60, −30 and 0 min before meal ingestion. Biopsy samples from the M. vastus lateralis were obtained at t = −180 and 0 min following initiation of the tracer infusion in order to assess basal postabsorptive muscle protein synthesis rates. Immediately following the second muscle biopsy (t = 0 min), participants were seated at a table and ingested the mixed meal (meal consumption time ≤15 min). To minimize the disturbance of the steady-state plasma L-[*ring*-$^{13}C_6$]phenylalanine precursor pool following meal ingestion, the infusion speed was increased by ∼12% to 0.057 μmol/kg/min for the remainder of the postprandial period, after which a third muscle biopsy was taken to determine the postprandial muscle protein synthetic response (0–240 min). Arterialized blood samples were collected at t = 15, 30, 45, 60, 90, 120, 150, 180, 210 and 240 min into the postprandial period.

Microvascular perfusion in the M. vastus lateralis was assessed at rest (at t = −90 min) and 60 min following meal ingestion (t = 60 min) by contrast-enhanced ultrasound (CEUS). All CEUS measurements were performed using a high-end ultrasound machine (Affinity 70G, Philips) with a linear array probe (eL18-4, Philips), as described previously (Betz et al., 2024). The ultrasound probe was fixed in a custom-made holder to visualize a cross-sectional image of the M. vastus lateralis at one-third distance between the superior patellar border and the anterior superior iliac spine. This position was marked on the leg by pen and a screenshot in ultrasound B-mode was taken to ensure repeated measurements at the exact same position. A 5 s clip was acquired in colour Doppler mode to be used during subsequent analysis. Next, an infusion of gas-filled microbubbles (SonoVue, Bracco, Milan, Italy; concentration: 8 μl/ml) was initiated via the catheter placed in the antecubital vein. For each CEUS measurement, a 10 ml suspension of microbubbles was infused for 6 min at 85 ml/h. Following 3 min of infusion to achieve a steady state of circulating microbubbles, six 30 s recordings were acquired using contrast mode (8 Hz with a mechanical index (MI) of 0.07). At the start of each recording, a high MI flash (0.53 MI) was given to destroy all visible microbubbles and the subsequent replenishment of microbubbles was recorded. Immediately following CEUS, femoral artery diameter and a 30 s average velocity were assessed using the same high-end ultrasound machine (Affinity 70G) with a linear array probe (eL18-4). Femoral artery blood flow was subsequently determined using the system's software (formula: blood flow = [π × (femoral artery radius)$^2$ × (mean blood flow velocity) × 60]). Systolic and diastolic blood pressure was determined to

calculate mean arterial pressure (formula: mean arterial pressure = {diastolic blood pressure + [1/3 × (systolic blood pressure – diastolic blood pressure)]}), and calculate femoral vascular conductance (ratio of blood flow/mean arterial pressure).

Blood samples were collected into EDTA-containing tubes and centrifuged at 1200 *g* for 10 min at 4°C. Aliquots of plasma were frozen in liquid nitrogen and stored at −80°C. Biopsy samples were collected with the use of a 5 mm Bergström needle (Bergstrom, 1975) custom-adapted for manual suction. Samples were obtained from separate incisions from the middle region of the M. vastus lateralis, ∼15 cm above the patella and ∼3 cm below entry through the fascia. Local anaesthetic (1% xylocaine with adrenaline 1:100,000) was applied to numb the skin and fascia. The first two muscle samples were taken from the same leg, and the last muscle biopsy was taken from the contralateral leg. Muscle samples were dissected carefully and freed from any visible non-muscle material. From fasted, postabsorptive muscle biopsies, we embedded a small part in Tissue-Tek (Sakura Finetek, Zoeterwoude, the Netherlands) to be frozen in liquid nitrogen-cooled isopentane for immuno-histochemical analyses. In addition, we sectioned a small portion of muscle (∼10 mg) and immediately placed it into ice-cold BIOPS preservation buffer for mitochondrial respiration experiments, as previously described (Petrick et al., 2019). All remaining muscle tissue was immediately frozen in liquid nitrogen. Muscle samples were subsequently stored at −80°C until further processing. When the experimental protocol was complete, cannulae were removed and participants were fed and assessed for ∼30 min before leaving the laboratory.

### Passive heat treatment

Over an 8-week period, participants visited our laboratory 3 times per week for a supervised session in an infrared sauna cabin (HM-LSE-3 Professional edition, Health Mate, Lier, Belgium). In order to acclimate to the PHT, the duration of the first PHT session was 35 min with a temperature at 50°C. This was followed, during the second session, by 40 min with a temperature at 55°C and, finally, 45 min with a temperature at 60°C for the remaining sessions (humidity is not controlled in an infrared sauna). At their own preference, participants were allowed to further raise the temperature. All sessions (including duration and temperature) were recorded by the investigators. Before and after each PHT session, body mass of each individual was measured. In addition, during each session, participants were allowed to drink water *ad libitum*. Following each PHT session, participants took a warm shower and were subsequently monitored for ∼30 min before leaving the laboratory.

### Meal composition and preparation

The food products used for the preparation of the meal were obtained from a local grocery store. The meal was composed of lean ground beef, potatoes, string beans (frozen, −20°C) and apple sauce (100% apples). Lean ground beef was obtained from the flank muscles of a 4-year-old, 520 kg, Belgian Blue cow from a local farm. The lean ground beef was vacuum sealed in single portions and stored at −20°C. The meals were prepared on the experimental test days and provided to the participants in thermal bowls.

Before the onset of the study, the meal composition was determined based on the macronutrient values for the prepared/cooked products, obtained from the Dutch Food Composition Database (NEVO, 2016). Meals were prepared according to a standard operating procedure. Cooking times were standardized and based on the recommended cooking time for each specific food product. Meat was cooked well-done in 7.5 min on medium temperature. A meat thermometer was used to confirm the meat temperature was 70–75°C.

To account for differences in body mass between individuals, participants were divided into body mass categories, in which the meal contents were scaled to ensure ∼0.35 g protein/kg body mass for different body mass ranges (i.e. 65–74 kg, 75–84 kg, 85–94 kg, etc.). As such, the meal provided 28 g protein for a 75–84 kg individual. In addition to protein content, carbohydrate, fat and total energy contents of the meals were also scaled to the body mass ranges, by adding or removing 12.5% compared to the standard 75–84 kg meal. For an individual of 75–84 kg, the meal contained 410 kcal with 28 g protein, 62 g carbohydrates, 3 g fat and 10 g fibre. For each individual, the exact same meal (i.e. composition and amount) was provided during the experimental test day before and after the 8-week passive heat treatment period. Meal ingestion time was recorded and did not differ before (11 ± 1 min) and after (9 ± 3 min) the 8-week passive heat treatment period ($P = 0.658$).

### Plasma analysis

Plasma glucose and insulin concentrations were analysed using commercially available kits (ref. no. A11A01667, Glucose HK CP, ABX Diagnostics, Montpellier, France; and ref. no. K151BZC, Meso Scale Discovery, Rockville, MD, USA, respectively).

Plasma amino acid concentrations were determined by ultra-performance liquid chromatography–mass spectrometry (UPLC-MS; Acquity UPLC H-Class with QDa; Waters, Saint-Quentin, France). Specifically, 50 μl blood plasma was deproteinized using 100 μl of 10% sulfosalicylic acid (SSA) with 50 μM of MSK-A2 internal standard (Cambridge Isotope Laboratories, Andover, MA,

USA). Subsequently, 50 μl of ultra-pure demineralized water was added and samples were centrifuged (15 min at 21,000 $g$). After centrifugation, 10 μl of supernatant was added to 70 μl of Borate reaction buffer (Waters). In addition, 20 μl of AccQ/Tag derivatizing reagent solution (Waters) was added after which the solution was heated to 55°C for 10 min. Of this 100 μl derivative 1 μl was injected and measured using UPLC-MS.

For the determination of plasma L-[*ring*-$^{13}C_6$]-phenylalanine enrichments, phenylalanine was derivatized to its 6-aminoquinolyl-*N*-hydroxysuccinimidyl carbamate (AQC) derivative, and enrichments were determined by UPLC-MS by using mass detection of masses 336, 342 and 346 for unlabelled and labelled $^{13}C_6$ and $^{13}C_9$ phenylalanine, respectively. Standard regression curves were applied from a series of known standard enrichment values against the measured values to assess the linearity of the mass spectrometer and to account for any isotope fractionation which may have occurred during the analysis.

### Skeletal muscle tissue analysis

**Muscle protein synthesis.** A piece of wet muscle (∼50–70 mg) was freeze dried for 48 h. Collagen, excessive blood and other non-muscle materials were subsequently removed from the muscle fibres under a light microscope. The isolated muscle fibre was weighed and 35 volumes (7× wet weight of isolated muscle fibres × wet-to-dry ratio 5:1) of ice-cold 2% perchloric acid (PCA) was added. Thereafter, the tissue was homogenized by sonication and centrifuged to separate the supernatant from the protein pellet. The protein pellet was washed 3 times with 1 ml 2% PCA. The amino acids were liberated from the mixed-muscle enriched protein fraction by adding 3 ml of 6 M HCl and heating to 110°C for 16 h. The hydrolysed mixed-muscle protein fractions were dried under a nitrogen stream while heated to 110°C. The dried mixed-muscle protein fraction was dissolved in a 50% acetic acid solution. The amino acids from the mixed-muscle protein fraction were passed over a Dowex exchange resin (AG 50W-X8, 100–200 mesh hydrogen form; Bio-Rad Laboratories, Hercules, CA, USA) using 2 M $NH_4OH$. Subsequently, the purified amino acid solution was dried under a nitrogen stream at room temperature, followed by derivatization to their *N*(*O*,*S*)-ethoxycarbonyl-ethylesters. The ratio of $^{13}C/^{12}C$ of mixed-muscle protein-bound phenylalanine was determined using gas chromatography–combustion-isotope ratio mass spectrometry (GC-IRMS; Delta V, Thermo Scientific, Bremen, Germany) by monitoring ion masses 44, 45 and 46. Standard regression curves were applied from a series of known standard enrichment values against the measured values to assess the linearity of the mass spectrometer and to account for any isotope fractionation which may have occurred during the analysis.

**Immunohistochemistry.** Frozen muscle biopsies were cut into 5-μm-thick cryosections using a cryostat at −20°C, and thaw-mounted onto uncoated pre-cleaned glass slides. Care was taken to properly align the samples for cross-sectional orientation of the muscle fibres. Muscle cross-sections were stained for muscle fibre type, laminin and capillaries. Following 5 min fixation in acetone and subsequent 15 min air drying at room temperature, the muscle cross-sections were incubated for 45 min with the first primary antibody CD31 (dilution 1/50; M0823; Dako, Glostrup, Denmark) in 0.05% Tween–phosphate-buffered saline (PBS). Slides were washed 3 × 5 min with PBS. Next, slides were incubated for 45 min in HAM Biotine (1:500, Vector Laboratories, Burlingame, CA, USA) in 0.05% Tween/PBS. Slide were again washed 3 × 5 min with PBS, followed by a 45 min incubation with Avidine Texas Red (A2006, dilution 1/400; Vector Laboratories) and antibodies against myosin heavy chain (MHC)-I (BA-F8, dilution 1/10; Developmental Studies Hybridoma Bank (DSHB)), and laminin (polyclonal rabbit anti-laminin, dilution 1/50; Sigma-Aldrich) in 0.05% Tween/PBS. Following a 3 × 5 min washing step in PBS, samples were finally incubated for 30 min with the appropriate secondary antibodies goat anti-mouse (GAM) IgG2b AlexaFluor488 and goat anti-rabbit (GAR) IgG AlexaFluor647 (Thermo Fisher Scientific, Waltham, MA, USA). After the final washing (3 × 5 min PBS), slides were mounted with Mowiol (Calbiochem, Fisher Scientific, Waltham, MA, USA). The staining procedure resulted in images with laminin in white, MHC-I in green, and CD31 in red (Fig. 4*A*).

Slides were viewed and automatically captured using a ×20 objective on a modified Olympus BX51 fluorescence microscope with a customized spinning disk unit (DSU, Olympus), computer-controlled excitation and emission filter wheels (Olympus, Tokyo, Japan), 3-axis high-accuracy computer-controlled stepping motor specimen stage (Grid Encoded Stage, Ludl Electronic Products, Hawthorne, NY, USA), ultra-high sensitivity monochrome electron multiplier CCD camera (C9100-02, Hamamatsu Photonics, Hamamatsu City, Japan) and controlling software (StereoInvestigator; MBF BioScience, Williston, VT, USA). Quantitative analyses were performed using ImageJ version 2.1.0/1.53c. On average, 330 ± 151 (before passive heat treatment) and 325 ± 119 (after passive heat treatment) muscle fibres were analysed per participant to determine muscle fibre type distribution and size. The quantification of muscle fibre capillaries was performed on 31 ± 2 type I and 31 ± 2 type II muscle fibres (before passive heat treatment)

and 31 ± 2 type I and 31 ± 5 type II muscle fibres (after passive heat treatment) based on the work of Hepple (1997). Quantification was made of (i) capillary contacts (CC), (ii) the capillary-to-fibre ratio (C/F*i*), (iii) capillary-to-fibre perimeter exchange (CFPE) index, and (iv) capillary density (CD). All immunofluorescence analyses were completed in a blinded fashion.

**Mitochondrial respiration.** For mitochondrial respiration experiments, muscle was trimmed of any fat and connective tissue under a microscope and separated into bundles using fine-tipped forceps. Bundles were permeabilized (40 μg/ml saponin in BIOPS) for 30 min while continuously rotating at 4°C (Petrick et al., 2019), followed by 15 min of washing in MiR05 respiration buffer (Petrick et al., 2020). Mitochondrial respiration experiments were subsequently performed at 37°C in 2 ml of MiR05 with constant stirring at 750 rpm, in an Oroboros Oxygraph-2k (Oroboros Instruments, Innsbruck, Austria). In the presence of 5 μM blebbistatin, 5 mM pyruvate and 1 mM malate, sequential titrations of ADP (25–10,000 μM ADP) were performed. Ten millimolar glutamate (maximal complex I-linked respiration), 10 mM succinate (maximal complex I+II-linked respiration) and 10 μM cytochrome *c* (to assess membrane integrity) were sequentially added. Any experiment in which cytochrome *c* increased respiration by more than 10% was excluded from analysis. Respiratory control ratios were calculated as: maximal ADP-supported respiration (presence of ADP)/pyruvate+malate-supported respiration (absence of ADP). All fibre bundles were recovered, freeze-dried, weighed and respiration data were normalized to fibre dry weight. Mitochondrial ADP kinetics were analysed using constrained one-phase associations ($Y_0 = 0$ and plateau = 100) to determine the apparent half-time using GraphPad Prism 9.0 (GraphPad Software, Boston, MA, USA).

**Mitochondrial protein content.** Freeze-dried muscle fibres were digested in lysis buffer (10% glycerol, 5% $\beta$-mercaptoethanol, 2.3% SDS, 62.5 mM Tris–HCl pH 6.8, with 0.005% bromophenol blue) at a ratio of 1 μl lysis buffer/5 μg dry fibre weight (Petrick et al., 2019). Fibres were digested for 1 h at 65°C with gentle shaking and were vortexed briefly every 15 min to improve digestion. Digested lysate was then loaded onto SDS-polyacrylamide gels (Criterion TGX Any kD, Bio-Rad). Five microlitres of lysate was loaded for OXPHOS and 10 μl for pyruvate dehydrogenase (PDH). Proteins were transferred onto a Trans-blot Turbo 0.2 μM nitrocellulose membrane (Bio-Rad). Commercially available antibodies were used to detected OXPHOS (1:500, Abcam, Cambridge, UK, Ab110411) and total PDH-E1$\alpha$ subunit (1:1000, Thermo

Fisher Scientific 459400). The samples were loaded onto two different gels, with pre- and post-samples for each participant always being loaded on the same gel. Both gels were subsequently transferred onto the same membrane to limit variability. Detection and quantification was performed using an Odyssey Infrared Imaging System (Li-Cor Biosciences, Lincoln, NE, USA).

## Microvascular perfusion analysis

All CEUS recordings were analysed using ImageJ (version 2.1.0/1.53c), as described previously (Betz et al., 2024). First, the colour Doppler clip was used to create a region of interest (ROI) that excludes larger blood vessels and connective tissue. This ROI was then used for the 30-s microbubble replenishment recording to determine video intensity for every frame. A background correction was applied by subtracting the average video intensity of the first 4 frames (i.e. 0.5 s) following the high MI flash from all data points. Video intensity data were plotted in GraphPad Prism (version 8.3) and a curve was fitted to the equation: $y = A(1 - e^{-\beta t})$, where $A$ is the plateau video intensity (i.e. microvascular blood volume) and $\beta$ reflects the rate of rise of video intensity (i.e. microvascular blood velocity) (Wei et al., 1998). Microvascular blood flow was calculated as the product of volume and velocity. For one CEUS measurement, the average of the six 30-s recordings were taken for microvascular blood volume, velocity and flow. All microvascular perfusion analyses were completed in a blinded fashion.

## Calculations

The fractional synthetic rates (FSR; in %/h) of mixed-muscle protein-enriched fractions was calculated by the standard precursor–product equation:

$$\text{FSR} = \left( \frac{E_{b2} - E_{b1}}{E_{\text{precursor}} \times t} \right) \times 100$$

where $E_{b2} - E_{b1}$ is the increment in mixed-muscle protein bound L-[*ring*-$^{13}C_6$]phenylalanine enrichment (mole percent excess) during the tracer incorporation period, and $t$ is the tracer incorporation time in hours. Weighted mean plasma enrichments were calculated by taking the measured enrichment between consecutive time points and correcting for the time between these sampling time points ($E_{\text{precursor}}$). For the calculation of basal FSR, $E_{b2}$ represented the protein bound L-[*ring*-$^{13}C_6$]phenylalanine enrichments in the muscle at $t = 0$ min, and $E_{b1}$ represented the protein-bound L-[*ring*-$^{13}C_6$]phenylalanine enrichments in the muscle at $t = -180$ min. For calculation for postprandial FSR, biopsy samples at $t = 0$ and 240 min were used.

Net incremental area under curve (iAUC) was determined for plasma glucose and insulin concentrations during the 4 h postprandial period following meal ingestion. The iAUC was calculated using the trapezoid rule, with plasma concentrations before meal ingestion ($t = 0$ min) serving as baseline.

### Statistical analysis

All data in text and tables are expressed as means ± SD. All data in figures are expressed as mean with individual data points or 95% confidence interval (CI), unless otherwise stated. Normal distribution of all parameters was verified by the Shapiro–Wilk test. Habitual physical activity level (week 1 *vs*. week 4 *vs*. week 8) assessed with the SQUASH questionnaire was determined with a one-factor repeated measures ANOVA. Plasma glucose, insulin, amino acid concentrations and enrichments were analysed by a two-factor (with time and treatment as within-subject factors) repeated measures ANOVA. In cases of significant main effects of time or treatment, Bonferroni-corrected *post hoc* analyses were performed to locate the differences. All other data were analysed with Student's paired-samples *t* test. Statistical significance was accepted as $P \leq 0.05$. All calculations were performed using SPSS (version 27.0, IBM Corp., Armonk, NY, USA).

### Results

#### Passive heat treatment

The total duration and average temperature of all PHT sessions were 45 ± 1 min and 63 ± 4°C, respectively. Body mass averaged 73.5 ± 12.0 kg before and 73.3 ± 11.9 kg after each of the PHT sessions. Participants ingested an average of 179 ± 190 ml of water during each PHT

**Table 1. Habitual dietary intake before (week 0) and at week 4 of the passive heat treatment (PHT) period, in healthy, older adults (n = 14; 9 males, 5 females)**

|  | Week 0 | Week 4 | P |
|---|---|---|---|
| Energy (kJ/day) | 8535 ± 2204 | 8896 ± 3938 | 0.495 |
| Energy (kcal/day) | 2074 ± 511 | 2207 ± 819 | 0.341 |
| Energy (kcal/kg BM/day) | 29 ± 9 | 31 ± 12 | 0.354 |
| Carbohydrate (g/day) | 238 ± 78 | 254 ± 107 | 0.493 |
| Carbohydrate (En%/day) | 45 ± 10 | 46 ± 11 | 0.929 |
| Fat (g/day) | 77 ± 24 | 78 ± 43 | 0.631 |
| Fat (En%/day) | 33 ± 9 | 31 ± 9 | 0.400 |
| Protein (g/day) | 98 ± 28 | 85 ± 42 | 0.432 |
| Protein (En%/day) | 19 ± 3 | 17 ± 4 | **0.048** |
| Protein (g/kg BM/day) | 1.4 ± 0.4 | 1.2 ± 0.6 | 0.439 |

Data are expressed as means ± SD. *P*-values shown in bold indicate statistical significance. BM, body mass. En%/day, energy percentage per day.

session. None of the participants reported any issues with the applied sauna protocol (attendance was 99 ± 1%).

#### Habitual dietary intake and physical activity

Dietary intake during the PHT period showed only a minor change for total protein energy percentage per day (En%/day), with all other variables not showing any differences (Table 1). For habitual physical activity, only light intensity activity level and duration had been reduced over the 8-week PHT period. Total activity level did not change over the 8-week period (Table 2).

**Table 2. Habitual physical activity level at week 1, 4 and 8 of the passive heat treatment (PHT) period, in healthy, older adults (n = 14; 9 males, 5 females)**

|  | Week 1 | Week 4 | Week 8 | P |
|---|---|---|---|---|
| Activity score (total min/week) | 6081 ± 4315 | 5545 ± 5633 | 5075 ± 3544 | 0.723 |
| Activity duration (min/week) | 2092 ± 1090 | 1759 ± 1386 | 1650 ± 936 | 0.194 |
| Light intensity activity score (total min/week) | 1730 ± 962 | 1276 ± 969 | 1141 ± 894* | **0.003** |
| Moderate intensity activity score (total min/week) | 3820 ± 3565 | 3861 ± 4746 | 3673 ± 3211 | 0.987 |
| High intensity activity score (total min/week) | 531 ± 1731 | 407 ± 1273 | 261 ± 747 | 0.336 |
| Light intensity activity duration (min/week) | 1300 ± 636 | 972 ± 764 | 903 ± 704* | **0.010** |
| Moderate intensity activity duration (min/week) | 732 ± 647 | 740 ± 843 | 717 ± 616 | 0.995 |
| High intensity activity duration (min/week) | 60 ± 193 | 47 ± 145 | 30 ± 84 | 0.336 |

Data are expressed as means ± SD. *P*-values shown in bold indicate statistical significance. *Significantly different compared to week 1 (*P*<0.010).

**Table 3. The effect of 8-week passive heat treatment (PHT) on body composition, muscle strength, physical function, and cardio-vascular health in healthy, older adults (*n* = 14; 9 males, 5 females)**

| | Pre PHT | Post PHT | *P* |
|---|---|---|---|
| Height (m) | $1.73 \pm 0.10$ | | |
| Body mass (kg) | $73.4 \pm 11.7$ | $73.1 \pm 12.0$ | 0.527 |
| BMI (kg/m$^2$) | $24.5 \pm 3.1$ | $24.5 \pm 3.0$ | 0.695 |
| Body lean mass (kg) | $50.2 \pm 9.3$ | $50.0 \pm 9.6$ | 0.606 |
| Body fat mass (kg) | $21.4 \pm 5.5$ | $21.2 \pm 5.4$ | 0.298 |
| Fat (%) | $29.0 \pm 6.3$ | $28.8 \pm 6.3$ | 0.657 |
| Bone mineral density (g/cm$^2$) | $1.13 \pm 0.11$ | $1.12 \pm 0.10$ | 0.572 |
| M. vastus lateralis CSA (cm$^2$) | $15.3 \pm 4.6$ | $15.2 \pm 4.6$ | 0.768 |
| Handgrip strength (kg) | $33 \pm 9$ | $35 \pm 11$ | **0.050** |
| 1RM leg press (kg) | $102 \pm 28$ | $107 \pm 31$ | 0.121 |
| 1RM leg extension (kg) | $46 \pm 12$ | $47 \pm 12$ | 0.087 |
| Gait speed (m/s) | $1.15 \pm 0.16$ | $1.24 \pm 0.20$ | 0.081 |
| Timed Up and Go (s) | $8.97 \pm 1.35$ | $8.55 \pm 1.13$ | 0.140 |
| Resting heart rate (bpm) | $60 \pm 7$ | $62 \pm 7$ | 0.829 |
| Systolic blood pressure (mmHg) | $139 \pm 10$ | $136 \pm 13$ | **0.019** |
| Diastolic blood pressure (mmHg) | $80 \pm 8$ | $77 \pm 10$ | 0.136 |

Data are expressed as means $\pm$ SD. *P*-values shown in bold indicate statistical significance. 1RM, one-repetition maximum; CSA, cross-sectional area.

### Body composition, muscle strength and function

Body mass, BMI, body lean mass, body fat mass, fat percentage and bone mineral density remained unchanged in response to 8 weeks of PHT (Table 3). In addition, no changes were observed for M. vastus lateralis CSA or leg muscle strength over time (Table 3). Handgrip strength was significantly increased following the intervention period ($P = 0.050$, Table 3). Gait speed and Timed Up and Go did not change in response to 8 weeks of PHT. Whereas diastolic blood pressure remained unchanged, average systolic blood pressure was significantly reduced (by 2%; $P = 0.019$) following 8 weeks of PHT. Resting heart rate did not change after 8 weeks of PHT (Table 3).

### Plasma glucose, insulin and amino acid concentrations

For postprandial plasma glucose concentrations, a main effect of time was observed ($P < 0.001$), with plasma glucose concentrations being elevated (from $t = 0$) between 30 and 180 min ($P < 0.05$). In addition, a significant treatment effect was observed ($P = 0.022$), showing lower postprandial plasma glucose concentrations following 8 weeks of PHT at time points $t = 15$ and 45 min ($P < 0.05$). Plasma glucose iAUC was significantly lower over the postprandial period following 8 weeks of PHT ($P = 0.024$; Fig. 2*A*). For postprandial plasma insulin responses, only a significant time effect was observed ($P < 0.001$), with plasma insulin concentrations being elevated (from $t = 0$) between 30 and 120 min ($P < 0.05$). No significant difference in postprandial

plasma insulin iAUC was observed between pre- and post-8 weeks of PHT ($P = 0.666$; Fig. 2*B*).

For postprandial plasma essential amino acid (Fig. 3*A*), non-essential amino acid (Fig. 3*B*) and total amino acid (Fig. 3*C*) responses, a significant time effect was observed ($P < 0.01$), with no significant treatment or interaction effects.

### Femoral artery responses

For femoral arterial diameter, femoral arterial blood velocity and femoral arterial blood flow no significant changes were observed during either the basal (post-absorptive) period or the postprandial period (at 60 min post-meal ingestion), when comparing before and after 8 weeks of PHT (Table 4). In addition, during the basal (postabsorptive) period no significant changes were observed for mean arterial pressure and femoral vascular conductance following 8 weeks of PHT. Post-prandial mean arterial pressure was significantly lower and femoral vascular conductance significantly higher following 8 weeks of PHT (Table 4).

### Muscle fibre characteristics and capillarization

For muscle fibre size and type distribution no significant changes were observed following 8 weeks of PHT (Table 5). A significant increase was observed for type I and type II muscle fibre CC, C/F*i* and CD in response to 8 weeks of PHT (Table 5). Furthermore, we observed a significant increase in the capillary-to-fibre perimeter

exchange index for both type I (from 4.60 ± 1.07 to 5.90 ± 0.92; $P < 0.001$) and type II (from 3.60 ± 0.86 to 4.62 ± 0.78; $P < 0.001$) muscle fibres over time (Fig. 4*B*).

### Muscle microvascular perfusion kinetics

Muscle microvascular perfusion kinetics are shown in Fig. 5. In the basal (postabsorptive) period, microvascular blood volume tended ($P = 0.093$) to be lower following 8 weeks of PHT. During the postprandial period, micro-vascular blood volume was significantly lower following 8 weeks of PHT ($P = 0.013$; Fig. 5*Ai*). In response to the meal ingestion, the microvascular blood volume response (expressed as fold change from basal) remained unchanged in response to 8 weeks of PHT ($P = 0.155$; Fig. 5*Aii*). Whereas a tendency ($P = 0.077$) was observed for basal (postabsorptive), postprandial microvascular blood velocity was significantly (∼2.5-fold; $P = 0.003$) higher following 8 weeks of PHT (Fig. 5*Bi*). In response to the meal ingestion, the microvascular blood velocity response (expressed as fold change from basal) remained unchanged in response to 8 weeks of PHT ($P = 0.259$; Fig. 5*Bii*). For microvascular blood flow, no changes were observed for either the basal ($P = 0.266$) or the post-prandial ($P = 0.717$) period following 8 weeks of PHT (Fig. 5*Ci*). In response to the meal ingestion, the micro-vascular blood flow response (expressed as fold change from basal) remained unchanged in response to 8 weeks of PHT ($P = 0.633$; Fig. 5*Cii*).

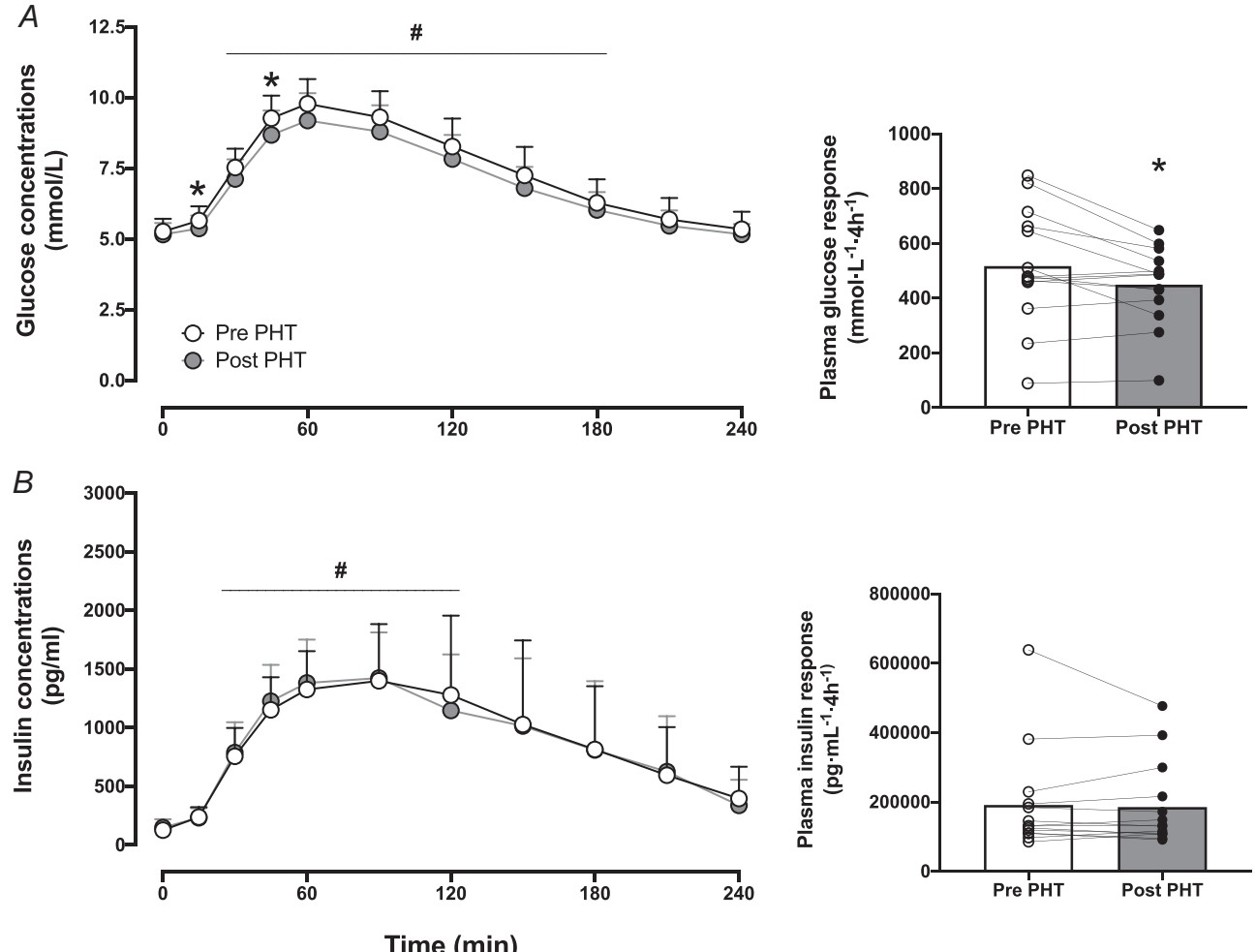

**Figure 2. Plasma glucose (*A*) and insulin (*B*) concentrations before (*t* = 0 min) and during 240 min following meal ingestion, before and after 8 weeks of passive heat treatment (PHT) in healthy, older adults (*n* = 14; 9 males, 5 females)**
The meal was ingested immediately after blood sampling at *t* = 0 min. In addition, the postprandial iAUC for both plasma glucose and insulin before and after 8 weeks of PHT is shown. Values represent means + 95% CI or individual values. *Significantly different ($P < 0.05$) from Pre PHT. #Significantly different compared with *t* = 0 min ($P < 0.05$).

**Table 4. The effect of 8 weeks of passive heat treatment (PHT) on femoral artery responses during the basal (postabsorptive) period and the postprandial period (at 60 min post meal ingestion) in healthy, older adults**

|  | Basal | | | Postprandial | | |
|---|---|---|---|---|---|---|
|  | Pre PHT | Post PHT | P | Pre PHT | Post PHT | P |
| Femoral arterial diameter (mm) | 6.5 ± 0.6 | 6.3 ± 1.2 | 0.401 | 6.4 ± 0.7 | 6.3 ± 1.2 | 0.477 |
| Femoral arterial blood velocity (cm/s) | 2.5 ± 2.1 | 2.9 ± 2.5 | 0.392 | 3.0 ± 1.7 | 3.9 ± 1.5 | 0.121 |
| Femoral arterial blood flow (ml/min) | 49.0 ± 43.6 | 51.1 ± 41.1 | 0.799 | 59.0 ± 34.4 | 71.5 ± 36.5 | 0.171 |
| Mean arterial pressure (mmHg) | 101 ± 11 | 100 ± 11 | 0.709 | 94 ± 11 | 88 ± 8 | **0.020** |
| Femoral vascular conductance (ml/min/mmHg) | 0.48 ± 0.51 | 0.53 ± 0.47 | 0.581 | 0.62 ± 0.39 | 0.85 ± 0.36 | **0.010** |

Data are expressed as means ± SD. *P*-values shown in bold indicate statistical significance. Given operational and technical issues: for femoral arterial diameter, blood velocity, blood flow, *n* = 11 (7 males, 4 females); and for mean arterial pressure and femoral vascular conductance, *n* = 10 (7 males, 3 females).

**Table 5. The effect of 8 weeks of passive heat treatment (PHT) on type I and type II muscle fibre characteristics and capillarization in healthy, older adults (*n* = 14; 9 males, 5 females)**

|  | Pre PHT | Post PHT | P |
|---|---|---|---|
| Muscle fibre size ($\mu m^2$) | | | |
| Type I | 5315 ± 1158 | 5399 ± 1370 | 0.619 |
| Type II | 3584 ± 1298 | 4098 ± 1084 | 0.336 |
| Fibre type (%) | | | |
| Type I | 49 ± 16 | 50 ± 15 | 0.886 |
| Type II | 51 ± 16 | 50 ± 15 | 0.886 |
| Capillary contacts | | | |
| Type I | 3.21 ± 0.73 | 3.91 ± 0.51 | **0.003** |
| Type II | 2.66 ± 0.55 | 3.21 ± 0.48 | **0.005** |
| C/F*i* | | | |
| Type I | 1.47 ± 0.37 | 1.90 ± 0.35 | **<0.001** |
| Type II | 0.96 ± 0.22 | 1.31 ± 0.23 | **<0.001** |
| CD (capillaries/$mm^2$) | | | |
| Type I | 305 ± 80 | 400 ± 93 | **<0.001** |
| Type II | 296 ± 100 | 371 ± 111 | **0.020** |

Data are expressed as means ± SD. *P*-values shown in bold indicate statistical significance. C/F*i*, individual muscle fibre capillary-to-fibre ratio; CD, capillary density.

## Muscle protein synthesis

For plasma [$^{13}C_6$]phenylalanine enrichments, no time ($P = 0.052$), treatment ($P = 0.677$) or interaction ($P = 0.579$) effects were observed (Fig. 6*A*). Meal ingestion did not elicit a change in muscle protein FSR either before ($P = 0.180$) or after ($P = 0.970$) 8 weeks of PHT (Fig. 6*B*). In addition, 8 weeks of PHT did not change FSR in the basal (postabsorptive; $P = 0.630$) or postprandial ($P = 0.199$) state.

## Mitochondrial protein content and respiratory function

Protein content of mitochondrial oxidative phosphorylation subunits (complexes I, II, III, IV, V) and mitochondrial pyruvate dehydrogenase (PDH) did not change following 8 weeks of PHT (Fig. 7*A* and *B*). In addition, PHT did not alter mitochondrial respiration in the absence of ADP (pyruvate+malate, PM), presence of saturating ADP, or presence of maximal complex I-linked (+glutamate) and I+II-linked (+succinate) substrates (Fig. 7*C*). The fold increase in respiration with ADP (respiratory control ratio; Fig. 7*C*) and sensitivity of mitochondria to ADP (Fig. 7*D*) did not change following 8 weeks of PHT.

## Discussion

The present study shows a substantial increase in muscle fibre capillarization following 8 weeks of PHT in older individuals. However, this did not lead to changes in skeletal muscle microvascular blood flow, basal and postprandial muscle protein synthesis rates, skeletal muscle (fibre) CSA, or leg muscle strength. Additionally, no changes were observed in muscle mitochondrial content and function after the PHT intervention. We did observe improvements in markers of cardiovascular health and postprandial glucose handling following 8 weeks of PHT in older individuals.

Whole-body PHT has been associated with many different health benefits, primarily through its ability to increase body temperature, heart rate, cardiac output and blood flow. Notably, consistent exposure to short periods of passive heat may result in beneficial alterations in skeletal muscle structure and function (Kim et al., 2020a).

In line with previous observations (Hesketh et al., 2019), we observed a substantial increase in both type I and type II skeletal muscle capillarization following the 8 weeks of PHT, without a shift in fibre type distribution or fibre size. Although this can be considered as a beneficial skeletal

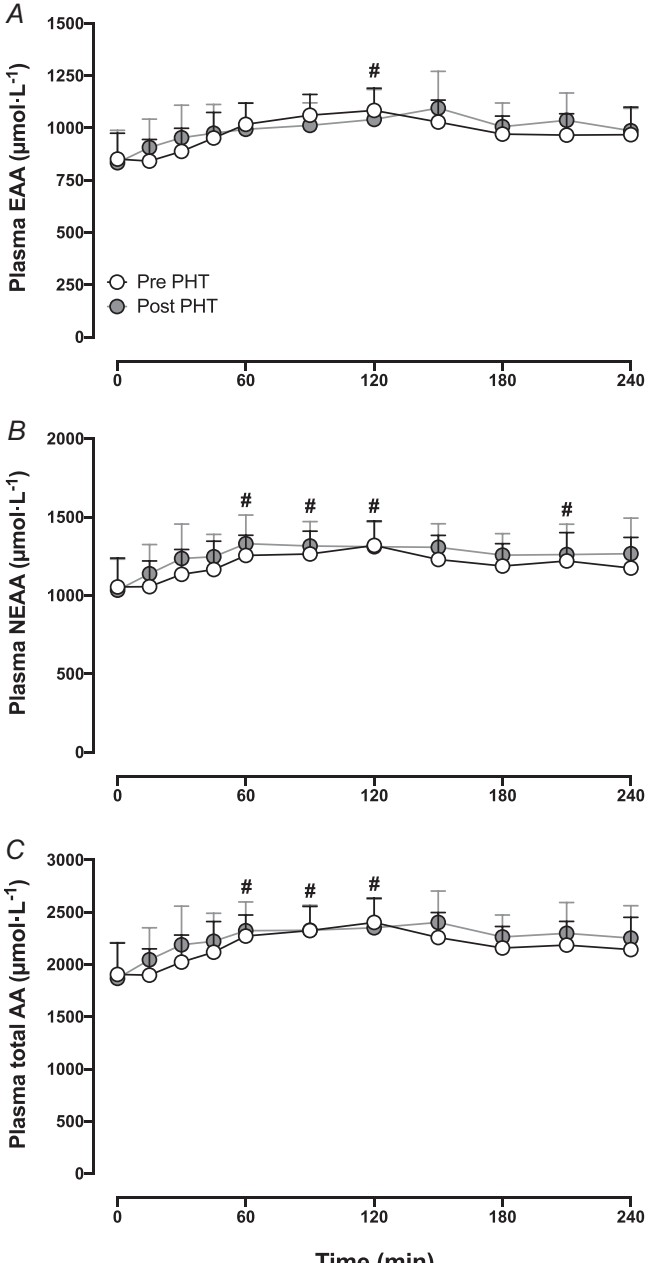

**Figure 3. Plasma essential amino acid (EAA; *A*), non-essential amino acid (NEAA; *B*) and total amino acid (AA; *C*) concentrations before (*t* = 0 min) and during 240 min following meal ingestion, before and after 8 weeks of passive heat treatment (PHT) in healthy older adults (*n* = 14; 9 males, 5 females)**
The meal was ingested immediately after blood sampling at *t* = 0 min. Values represent means + 95% CI. #Significantly different compared with *t* = 0 min (*P* < 0.05).

muscle adaptation, it is important to recognize that muscle fibre capillarization is a static measurement and shows anatomical perfusion potential, but does not reflect actual perfusion dynamics. Therefore, we included CEUS measurements to assess muscle microvascular perfusion dynamics in the fasted state and at 60 min after mixed meal ingestion, before and after 8 weeks of PHT. Surprisingly, no differences in postabsorptive and postprandial muscle microvascular blood flow were observed in response to the 8-week PHT intervention. The absence of changes in postabsorptive or postprandial muscle microvascular blood flow, despite the considerable increase in muscle fibre capillarization, implies that the capillary network is not restrictive for basal or postprandial muscle microvascular perfusion in older individuals. Whether the PHT-induced increase in muscle fibre capillarization allows greater microvascular blood flow under conditions of greater blood flow requirements, such as during and/or following (sub)maximal exercise, remains to be established. This may be of particular relevance in older adults, as we have previously observed a strong association between type II muscle fibre capillarization and postexercise microvascular blood volume in this population (Betz et al., 2024). In the present study, we assessed muscle microvascular blood flow responses before and 60 min after meal ingestion only, when insulin was projected to be at its peak. Although insulin was indeed observed to peak at around 60–90 min after meal ingestion, the transient nature of the response does not exclude the possibility that differences may have been present at other postprandial time points.

We hypothesized that PHT would increase muscle fibre capillarization and perfusion capacity, thereby increasing muscle protein synthesis rates in older individuals. In this study, postprandial muscle protein synthesis rates were assessed over 4 h following the ingestion of a mixed meal, providing ∼0.35 g of protein per kg body mass (group average: 26 ± 5 g protein). A similar quantity of ingested protein has been demonstrated to elevate circulating plasma amino acid levels and stimulate muscle protein synthesis rates in older individuals (Fuchs et al., 2019; Pennings et al., 2011). Consistent with this, we observed an increase in plasma amino acid concentrations, peaking at *t* = 60–120 min following meal ingestion. The rise in plasma amino acid concentrations was, however, modest when compared to other studies from our laboratory where there was ingestion of a similar amount of protein (Fuchs et al., 2019; Pennings et al., 2011). This is likely attributed to the fact that in the current study participants ingested wholefoods, and not a protein isolate or concentrate. Wholefoods are more slowly digested and absorbed, resulting in an attenuated postprandial rise in circulating amino acid concentrations (Hermans et al., 2022). Moreover, the mixed meal contained a small amount of fat as well as carbohydrates (∼0.8 g/kg body

mass), which have also been reported to delay *in vivo* protein digestion and amino acid absorption (Gorissen et al., 2014). The relatively modest increase in post-prandial plasma amino acid concentrations likely explains our inability to detect postprandial increases in muscle protein synthesis rates. More importantly, however, basal and/or postprandial muscle protein synthesis rates were not enhanced following 8 weeks of PHT. This finding underscores that a denser muscle fibre capillary network, resulting from 8 weeks of PHT, does not augment basal and/or postprandial muscle protein synthesis rates in older adults. Whether the increased muscle fibre capillarization may be of benefit to augment the muscle protein synthetic response to exercise remains to be established.

In line with the absence of changes in muscle protein synthesis rates, no changes in skeletal muscle size were observed in response to the 8 weeks of PHT, neither on a micro (muscle fibre CSA) nor macro (M. vastus lateralis CSA and DXA derived lean mass) level. These data further support the hypothesis that heat treatment does not enhance skeletal muscle anabolism *in vivo* in humans (Fuchs, Smeets et al., 2020; Kim et al., 2020b; Labidi et al., 2021; Stadnyk et al., 2018), as has been suggested previously (Goto et al., 2007; Goto et al., 2011). We also observed no changes in leg muscle strength, gait speed and TUG test in response 8 weeks of PHT. Interestingly, however, we did observe a small (∼ 6%),

but significant, improvement in handgrip strength over time. The exact mechanism of improved force-generating capacity, without apparent concomitant improvements in other physical function outcomes and/or lean tissue mass, remains unclear. Irrespective of the mechanism(s), increased handgrip strength may have a clear clinical significance for older individuals, given its associations with health status (Bohannon, 2019).

Acute whole-body heat exposure has been shown to enhance gene expression of several targets associated with skeletal muscle mitochondrial biogenesis (Ihsan et al., 2020). Whereas some have (Hafen et al., 2018; Hafen et al., 2019; Marchant et al., 2022), others have not (Hesketh et al., 2019; Kim et al., 2020b) observed enhanced muscle mitochondrial content and/or function in response to different longer-term PHT strategies. In the present study, we show that 8 weeks of repeated exposure (3× per week) to a strong whole-body heat stress (infrared sauna; 45 min at 63°C) did not change skeletal muscle mitochondrial function (i.e. mitochondrial respiration) or content. This discrepancy may be explained by the premise that the muscle may need to reach a certain temperature threshold (e.g. >39–40°C) to induce changes in muscle mitochondrial function and/or content (Marchant et al., 2023). Achieving such an increase in muscle temperature is likely more feasible with more intense localized heating modalities (i.e. pulse wave diathermy) as compared with whole-body PHT.

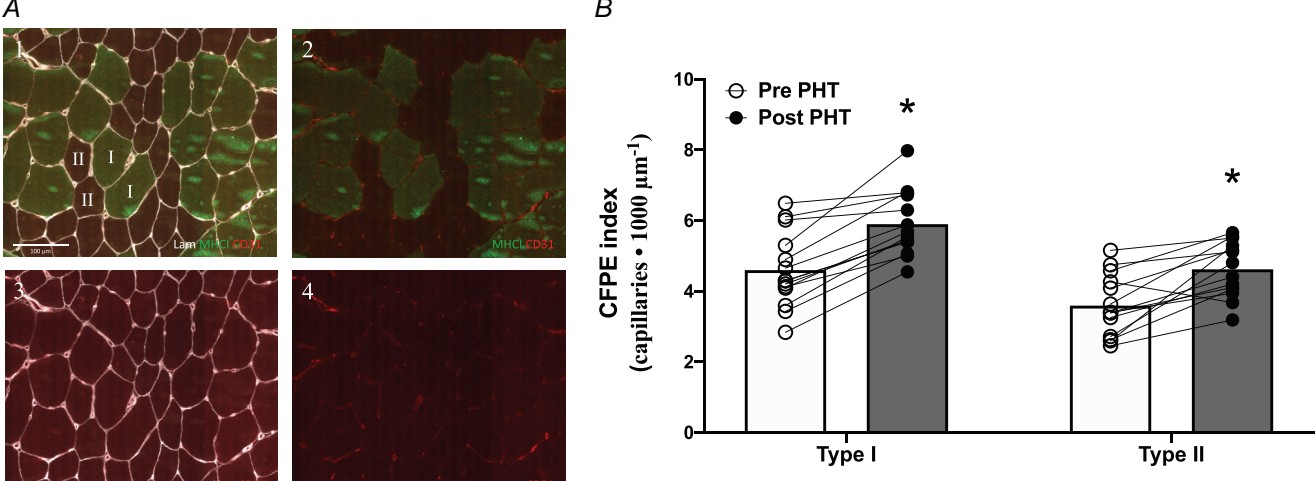

**Figure 4. Fibre type-specific microvascular staining (A) and capillary-to-fibre perimeter exchange (CFPE) index (B) before and after 8 weeks of passive heat treatment (PHT) in healthy older adults (*n* = 14; 9 males, 5 females)**
*A*, representation of fibre type-specific analysis of skeletal muscle microvascular staining. Type I muscle fibres are shown in green (MHCI), type II muscle fibres in black, cell walls in white (laminin) and capillaries in red (CD31). Scale bar, 100 μm. *B*, capillary-to-fibre perimeter exchange (CFPE) index in both type I and type II muscle fibres, before and after 8 weeks of passive heat treatment (PHT) in healthy, older adults (*n* = 14; 9 males, 5 females). Values represent means and individual values. *Significantly different (*P* < 0.001) from Pre PHT. [Colour figure can be viewed at wileyonlinelibrary.com]

Apart from skeletal muscle adaptations, the exposure to repeated heat stress by PHT may provoke beneficial haemodynamic and metabolic adaptations. Indeed, repeated passive heat exposure has been linked to reduced risk of fatal cardiovascular events and all-cause mortality (Laukkanen et al., 2015), which is attributed to improved cardiovascular health (Brunt & Minson, 2021; Brunt et al., 2016). In the present study, we show a significant reduction in systolic blood pressure as well as postprandial mean arterial pressure in response to 8 weeks of PHT, in healthy, older individuals. Furthermore, we observed an improved postprandial femoral vascular conductance following PHT. These improvements in vascular function and blood pressure are in line with other experimental studies evaluating the health benefits of repeated heat exposure interventions (Pizzey et al., 2021). Besides its

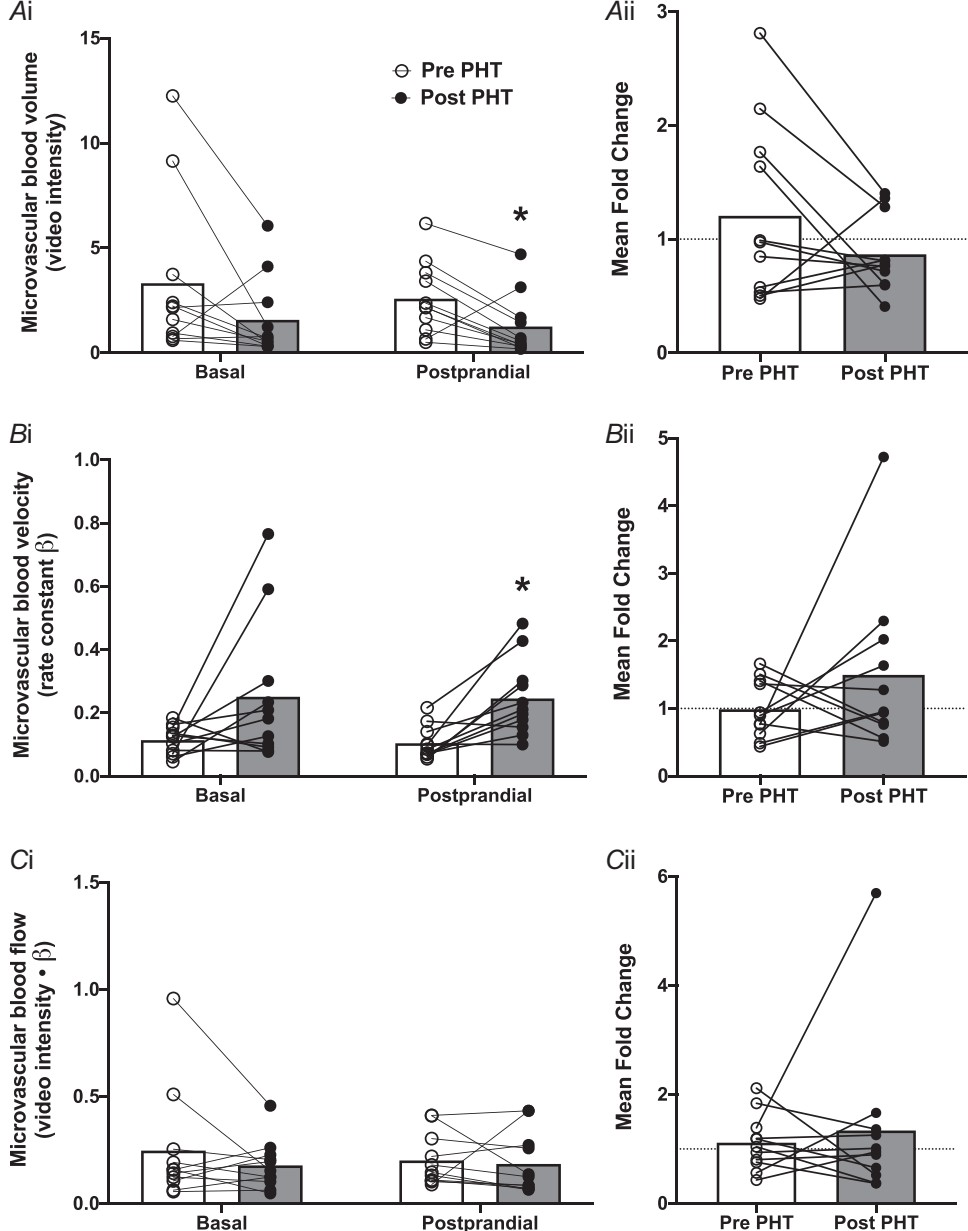

**Figure 5. Microvascular blood volume (*Ai*), velocity (*Bi*), and flow (*Ci*) in the basal (postabsorptive) and postprandial period, before and after 8 weeks of passive heat treatment (PHT) in healthy older adults**
In addition, the mean fold change (postprandial/basal) before and after 8 weeks of PHT is shown for microvascular blood volume (*Aii*), velocity (*Bii*), and flow (*Cii*). $n = 11$ (7 males, 4 females), due to operational and technical issues. Values represent means and individual values. *Significantly different ($P < 0.05$) from Pre PHT.

cardiovascular benefits, there is increasing evidence that prolonged PHT improves glucose metabolism. Fasting glucose and/or insulin concentrations have been reported to be lowered following 8–10 passive heat exposure sessions in overweight individuals (Hoekstra et al., 2018; Pallubinsky et al., 2020) and patients with type II diabetes mellitus (James et al., 2023). Furthermore, postprandial glucose and insulin concentrations (following 75 g of glucose ingestion) have been shown to be lower following 30 hot tub sessions (of 60 min at 40.5°C) over 8–10 weeks in obese women with polycystic ovary syndrome (Ely et al., 2019). In the present study, we assessed the impact of 8 weeks of whole-body PHT on postprandial glucose

and insulin responses following the intake of a mixed meal, offering a more practical and ecologically valid setting. Although no changes were observed in the postprandial insulin responses, glucose concentrations were significantly lower following meal ingestion after 8 weeks of PHT. Interestingly, the improved glycaemic control was particularly evident in participants who initially showed the highest postprandial plasma glucose response ($>500$ mmol/l/4 h). This provides further support for the suggestion that benefits of PHT for metabolic health may particularly be effective for those individuals with a more impaired glucose metabolism, like (pre)diabetic people (Hooper, 1999; James et al., 2023). Overall, the present findings confirm that prolonged whole-body PHT improves cardiometabolic health in older individuals. In the present study, the improved systolic blood pressure and/or postprandial glucose handling after the 8 weeks of PHT was, however, not correlated with the increase in type I and type II muscle fibre capillarization.

The present study is the first to provide an extensive evaluation on the skeletal muscle adaptive response to PHT in healthy older adults. Although the study included both healthy older males and females, the overall sample size was too small to assess sex-specific differences in the response to 8 weeks of PHT. Additionally, it could be speculated that the sample size was insufficient to detect a statistically significant change in postprandial muscle protein synthesis rates following PHT, as a moderate effect size ($d = 0.53$) was observed for this specific outcome parameter. Furthermore, it remains to be determined whether the lack of changes in basal and/or postprandial microvascular blood flow and muscle protein synthesis rates following 8 weeks of PHT would be similarly observed in younger and/or more clinically compromised individuals. Finally, it is important to note that the study did not include a non-intervention control group. However, the primary outcomes are unlikely to change significantly over a relatively short 8-week period in the absence of a specific intervention.

In conclusion, prolonged whole-body PHT enhances skeletal muscle fibre capillarization in healthy older adults. However, the denser muscle fibre capillary network does not improve basal or postprandial muscle microvascular blood flow or augment muscle protein synthesis rates or mitochondrial content and function. In addition, prolonged passive heat treatment does not increase skeletal muscle fibre size, lean tissue mass or leg muscle strength. However, prolonged whole-body PHT does improve markers of cardiovascular health and postprandial glucose handling.

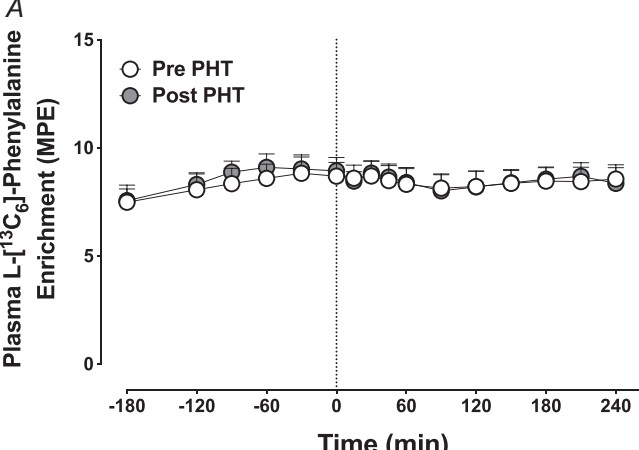

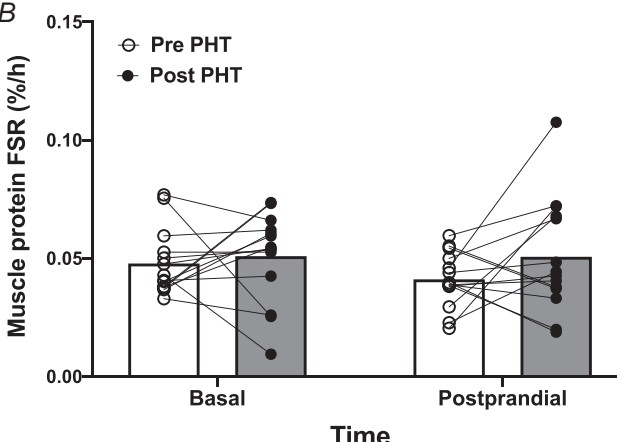

**Figure 6. Plasma L-[$^{13}$C$_6$]phenylalanine enrichments over time, before and after 8 weeks of passive heat treatment (PHT) (*A*) and muscle protein fractional synthetic rates (FSR) in the basal (postabsorptive) and postprandial period, before and after 8 weeks of PHT (*B*) in healthy older adults (*n* = 14; 9 males, 5 females)**
The dashed line (*A*) represents meal ingestion (*t* = 0 min). Values represent means + 95% CI (*A*) or individual values (*B*).

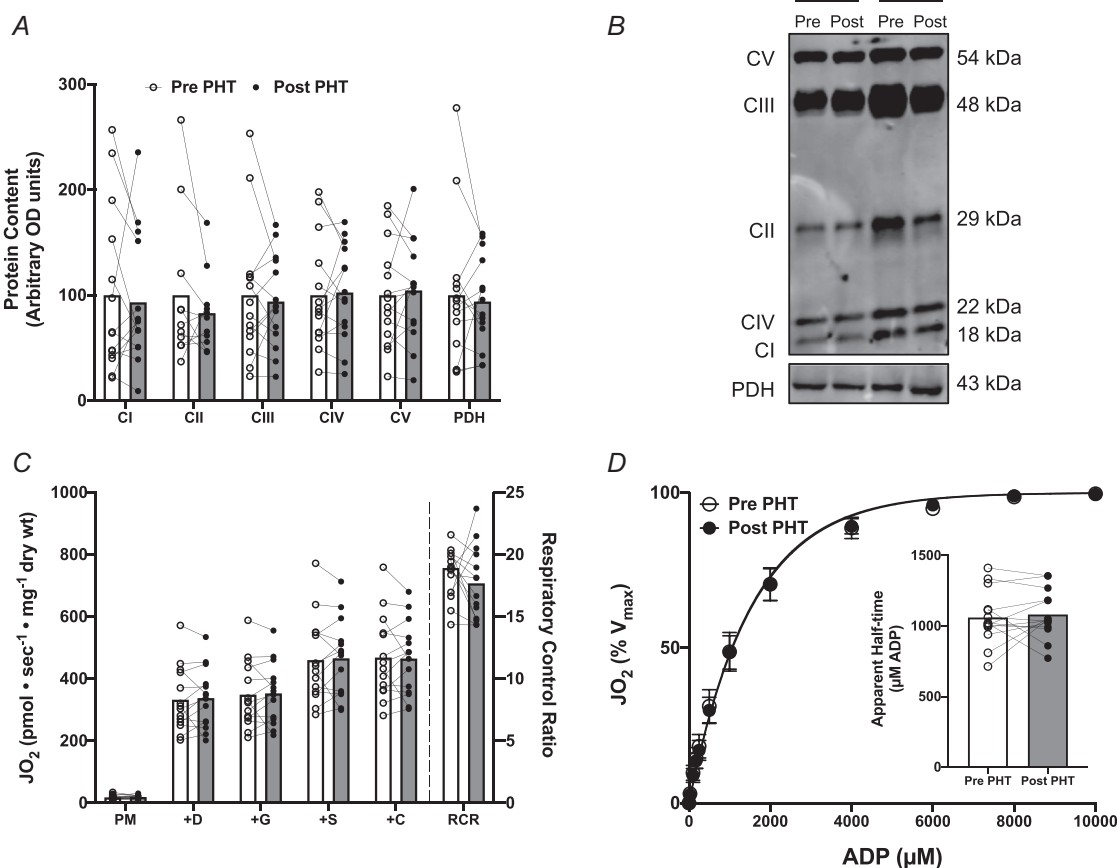

**Figure 7. Mitochondrial protein content (*A* and *B*), maximal mitochondrial respiratory capacity (*C*), and mitochondrial ADP sensitivity (*D*) in skeletal muscle before (pre) and after (post) 8 weeks of passive heat treatment (PHT)**

Data are expressed as means with individual responses superimposed, or means $\pm$ SD, where appropriate ($n = 14$; 9 males, 5 females). For OXPHOS CII, $n = 11$ (7 males, 4 females) due to artefacts on membrane preventing accurate quantification. ADP, adenosine diphosphate; +C, +cytochrome *c*; CI, OXPHOS complex I (NDUFB8); CII, OXPHOS complex II (SDHB), CIII, OXPHOS complex III (UQCRC2); CIV, OXPHOS complex IV (COX II); CV, OXPHOS complex V (ATP5A); +D, +ADP; +G, +glutamate; $JO_2$, oxygen consumption; OD, optical density; OXPHOS, oxidative phosphorylation system; PDH, pyruvate dehydrogenase (E1$\alpha$ subunit); PM, pyruvate+malate; RCR, respiratory control ratio; +S, +succinate.

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

## Additional information

### Data availability statement

All data generated and analysed during this study are available from the corresponding author on reasonable request.

### Competing interests

All authors declare no conflicts of interest related to this study.

### Author contributions

C.J.F., M.W.B., L.J.C.L. and T.S. conceived and designed the research. C.J.F., M.W.B., J.W., F.K.H., J.L.M.B. and T.S. generated and collected data. C.J.F., M.W.B., H.L.P., J.W., J.M.S. and T.S. analysed the data and/or interpreted the results. C.J.F., H.L.P. and T.S. prepared the figures. C.J.F. and T.S. wrote the manuscript. C.J.F., M.W.B., H.L.P., J.W., J.M.S., F.K.H., J.L.M.B., L.J.C.L. and T.S. revised the manuscript. All authors have read and approved the final version of this manuscript and agree to be accountable for all aspects of the work in ensuring that questions related to the accuracy or integrity of any part of the work are appropriately investigated and resolved. All persons designated as authors qualify for authorship, and all those who qualify for authorship are listed.

### Funding

This work was supported by ZonMw under the Offroad program, grant number: 04510012010023.

### Acknowledgements

The authors would like to thank Andrew M. Holwerda for helping with preparations of the mixed meals during the trials, Alfons J. H. M. Houben for useful discussions, and Wendy Sluijsmans, Hasibe Aydeniz, Annemie P. Gijsen and Joy P. B. Goessens for their (technical) assistance. Furthermore, the authors would like to thank the participants for their enthusiastic support and to volunteer to participate in this experiment. Finally, we greatly appreciate and thank Health Mate for providing a HM-LSE-3 infrared sauna cabin to use in the present study.

### Keywords

adaptation, ageing, blood pressure, glucose metabolism, heat stress, hypertrophy, mitochondrial content, mitochondrial function, stable isotope tracers, vascular conductance

### Supporting information

Additional supporting information can be found online in the Supporting Information section at the end of the HTML view of the article. Supporting information files available:

**Peer Review History**

