## [Peer Review History · The Journal of Physiology]

Repeated passive heat treatment increases muscle tissue capillarization, but does not affect postprandial muscle protein synthesis rates in humans

Cas J Fuchs, Milan W Betz, Heather Petrick, Jil Weber, Joan Senden, Floris K Hendriks, Julia L.M. Bels, Luc JC van Loon, and Tim Snijders

DOI: 10.1113/JP286986

Corresponding author(s): Tim Snijders (tim.snijders@maastrichtuniversity.nl)

The following individual(s) involved in review of this submission have agreed to reveal their identity: Matthew Robinson (Referee #1)

Review Timeline:

Submission Date:	28-May-2024
Editorial Decision:	17-Jun-2024
Revision Received:	15-Aug-2024
Editorial Decision:	27-Aug-2024
Revision Received:	11-Sep-2024
Accepted:	13-Sep-2024

Senior Editor: Paul Greenhaff

Reviewing Editor: Christopher Sundberg

Transaction Report:

Dear Dr Snijders,

Re: JP-RP-2024-286986 "Repeated passive heat treatment increases muscle tissue capillarization, but does not affect postprandial muscle protein synthesis rates in humans" by Cas J Fuchs, Milan W Betz, Heather Petrick, Jil Weber, Joan Senden, Floris K Hendriks, Julia L.M. Bels, Luc JC van Loon, and Tim Snijders

Thank you for submitting your manuscript to The Journal of Physiology. It has been assessed by a Reviewing Editor and by 2 expert referee and we are pleased to tell you that it is potentially acceptable for publication following satisfactory major revision.

Please address all the points raised and incorporate all requested revisions or explain in your Response to Referees why a change has not been made. We hope you will find the comments helpful and that you will be able to return your revised manuscript within 2 months. If you require longer than this, please contact journal staff: jp@physoc.org. Please note that this letter does not constitute a guarantee for acceptance of your revised manuscript.

LANGUAGE EDITING AND SUPPORT FOR PUBLICATION: If you would like help with English language editing, or other article preparation support, Wiley Editing Services offers expert help, including English Language Editing, as well as translation, manuscript formatting, and figure formatting at www.wileyauthors.com/eoo/preparation. You can also find resources for Preparing Your Article for general guidance about writing and preparing your manuscript at www.wileyauthors.com/eoo/prepresources.

REVISION CHECKLIST:

We look forward to receiving your revised submission.

Yours sincerely,

Paul Greenhaff
Senior Editor
The Journal of Physiology

REQUIRED ITEMS

- Author photo and profile. First or joint first authors are asked to provide a short biography (no more than 100 words for one author or 150 words in total for joint first authors) and a portrait photograph. These should be uploaded and clearly labelled together in a Word document with the revised version of the manuscript. See Information for Authors for further details.

- The Journal of Physiology funds authors of provisionally accepted papers to use the premium BioRender site to create high resolution schematic figures. Follow this link and enter your details and the manuscript number to create and download figures. Upload these as the figure files for your revised submission. If you choose not to take up this offer, we require figures to be of similar quality and resolution. If you are opting out of this service to authors, state this in the Comments section on the Detailed Information page of the submission form. The link provided should only be used for the purposes of this submission. Authors will be charged for figures created on this premium BioRender account if they are not related to this manuscript submission.

- Please upload separate high-quality figure files via the submission form.

- You must upload original, uncropped western blot/gel images (including controls) if they are not included in the manuscript. This is to confirm that no inappropriate, unethical or misleading image manipulation has occurred. These should be uploaded as 'Supporting information for review process only'. Please label/highlight the original gels so that we can clearly see which sections/lanes have been used in the manuscript figures. For more information, see: <https://physoc.onlinelibrary.wiley.com/hub/journal-policies#imagmanip>.

EDITOR COMMENTS

Reviewing Editor:

Thank you for submitting your manuscript to The Journal of Physiology to be considered for inclusion in the special issue on the physiology of ageing skeletal muscle and the protective effects of exercise. Two expert reviewers evaluated your manuscript and while both were enthusiastic about the comprehensive approach and potential impact of the work, they both had several major concerns. Specifically, the introduction does not currently set the stage well for the state of the field on passive heat treatment and the different outcome variables that were studied. There is also concern that a young cohort or a time control group were not included in the experimental design and that the sample size may have resulted in being statistically underpowered for several of the outcomes.

Please also see 'Required Items' above.

Senior Editor:

Thank you for the manuscript submission to The Journal of Physiology (TJP), which has been considered by a reviewing editor and two expert reviewers. Overall the evaluation of the manuscript was positive and although the findings reported are essentially negative the broad-array of end point measurements convey an important message. Nevertheless, a number of major and minor concerns have been raised which the authors need consider and address on a point-by-point basis when revising the manuscript if they wish the manuscript to be considered further. Thank you for considering TJP and we look forward to receiving the revised manuscript and rebuttal.

REFeree COMMENTS

Referee #1:

The authors investigated if heating could be an intervention to improve anabolic response in older adults. They tested this possibility using a repeated measures design of anabolic response (measured by isotopic tracers), muscle capillarization and blood flow, and mitochondrial function. All of these measures have been hypothesized to be mechanisms by which heating may impact skeletal muscle, which is an important consideration in fundamental understanding of muscle physiology. The practical application of the findings is that heating did not change the muscle protein synthesis response in either basal (fasted) or in fed-state. There was an increase in capillarization, which was measured by multiple indexes, but no change in blood flow. This is an interesting finding that challenges a notion that greater capillaries drive blood flow and nutrient delivery.

The manuscript is overall very clear and methods described well. Strengths include the attention to providing a consistent meal, heating intervention and multiple complementary measurements. The discussion is overall clear integration of findings with the field. Major limitations in study design are lack of young (major) or non-heating group to account for time (minor, as it is not anticipated that there will be muscle change in parameters like synthesis rates over 8 weeks).

Major

- 1) Introduction: Introduction lacks strong rationale to study heating as an intervention. As written, the intro focuses on anabolic resistance of aging, which the study design does not test because it does not include a younger group (see point 2). Currently, heating comes in the final section of 3rd paragraph of the introduction, but is the first line of the abstract. The introduction should be revised to better establish the state of the field of heating on blood flow, angiogenesis and mitochondrial function.
- 2) Discussion: The discussion does not address limitation or context of the findings. Specifically, please address the lack of young control group and possible sex effects (such as on protein synthesis or capillarization, some reports both are higher in females).
- 3) Results: There are variable responses in blood flow and protein synthesis. Is there correlation between blood flow or capillaries with protein synthesis, particularly the changes? It could be that heating increased protein synthesis in those that also increase capillaries/blood flow, which would align with the hypothesis. I suggest running correlation between changes in blood flow and capillaries with protein synthesis.

Minor

- 4) Suggest to temper lines about "cardiovascular health" (such as 612) since these claims seem based on a decrease in systolic BP by 3 mmHg.
- 5)
- 6) Please include number of males and females in methods and figure legends.
- 7) Figures: The uses of "Ai and Aii" in figure 5 is not conventional. Please confirm with journal if this reporting aligns with journal standards.
- 8) Title: Suggest adding age to title, possibly as "...rates in older human"

9) Figures: Suggest adding an arrow or other indication for the meal at time 0.

10) Line 576: Remove paragraph break.

11) Clarify line 378: Do you mean all samples were loaded on same gel?

Referee #2:

Fuchs and colleagues present a study investigating infrared sauna therapy (passive heat treatment; PHT) on muscle capillarization and muscle protein synthesis in older adults. They build upon the evidence that age-related decline in muscle perfusion is associated with impaired muscle function and anabolism and that PHT improves perfusion. Thus, it is a logical "next step" to investigate the effect of PHT on the skeletal muscle hypertrophy. Importantly, Fuchs and colleagues investigate this question in older adults who may have impaired perfusion. The investigators found that, as expected, PHT improved muscle capillarization and blood pressure and femoral vascular conductance. However, the authors indicate there were no improvements in microvascular perfusion nor differences in the anabolic response to feeding. In all, this is a relevant investigation in the potential therapeutic potential of heat therapy.

While the study comprehensively assessed relevant physiological outcomes (blood flow, glucose/insulin concentrations, muscle FSR, and function), there is one concern regarding statistical power that should be noted.

While the field of science has a problem with biasing towards "positive" outcomes, another challenge is arriving to a false negative because inadequate power. Of course, it is difficult to anticipate the effect size of an intervention on every outcome of interest. The authors cite Timmerman et al. as evidence that perfusion to skeletal muscle can influence the anabolic response to insulin. Timmerman and colleagues infused sodium nitroprusside, a potent vasodilator in older adults along with insulin. They reported a robust increase in blood flow (Cohen's $d \sim 3.37$) and a concomitant increase in FSR (Cohen's $d \sim 2.76$). In the present study, the effect size of PHT on blood flow is moderate ($d = 0.52$) based on the reported data, which is 6 fold less than SNP. The effect size of PHT on FSR was, quite interestingly, also 0.52 (based on reported data). However, if we could assume that the effect of PHT on FSR would be similarly lower. As a result, doing a power calculation assuming an effect size of 0.50 would suggest one would need ~ 30 subjects to detect a statistically significant difference (alpha set to 0.05, powered to 0.80). Of course, this is all in hindsight. In many ways, one could consider this certainly longwinded comment as unproductive. However, this reviewer would suggest at least some commentary to be made to the fact that this is a small sample size given the intervention. Of course, the critical reader will recognize the sample size as a limitation and could likely draw their own conclusion. At the same time you are discussing the sample of the study, you could also highlight the strength of having older individuals in this study.

Despite this concern, the data presented seem to be well represented by the manuscript. Another weakness is a lack of a control group for the 8 weeks, but it is understandably difficult to include one given the time and expense.

Minor comments:

The reviewer does have concerns with the use of utilizing paired t-tests. For example, it seems more appropriate to conduct a Two-Way ANOVA on microvascular volume, velocity, and blood flow and on muscle FSR. To compare FSR both ways (i.e. pre/post PHT and pre/post feeding), it seems a Two-Way ANOVA would be necessary. Perhaps it would be worth checking with a statistician. However, it seems unlikely a different statistical approach would change conclusions.

The authors recruited both males and females, were there any patterns of sex-specific/dimorphic responses?

The decline in light physical activity is a surprise, is this due to the time travelling to/spent in the PHT, is the decrease in light

physical activity a consequence of fatigue from the sauna?

It is quite interesting that PHT seems to benefit those with [slightly] impaired glucose control post feeding (Figure 2B), was there any association between the relative change (%) in outcomes such as glucose and blood flow with the relative change in outcomes such as FSR, grip strength, or gait speed.

Lines 717 - 719: the authors state that the "improved systolic blood pressure and/or

postprandial glucose handling after the 8 weeks of PHT was, however, not associated with

the increase in type I and type II muscle fiber capillarization." However, the authors report that capillarization in myofibers were improved (Table 5 and Figure 4). Or are the authors referring to microvascular blood flow as they conclude that microvascular blood flow does not change (Figure 5 and Lines 625-627)?

Line 577 - 580: I'm confused. Do you mean: "Feeding did not elicit a change in muscle protein FSR either before ($P = 0.180$) or after ($P=0.970$) 8 weeks of PHT. In addition, 8 weeks of PHT did not change FSR in either the basal (postabsorptive) ($P=0.630$) or postprandial ($P=0.199$) state."

The authors included the FSR data in the abstract, but will they also considering including it in the results of the manuscript (Lines 574 - 580)?

END OF COMMENTS

Confidential Review

28-May-2024

Response to Referees

REQUIRED ITEMS

- Author photo and profile. First or joint first authors are asked to provide a short biography (no more than 100 words for one author or 150 words in total for joint first authors) and a portrait photograph. These should be uploaded and clearly labelled together in a Word document with the revised version of the manuscript. See Information for Authors for further details.

This has been added.

- The Journal of Physiology funds authors of provisionally accepted papers to use the premium BioRender site to create high resolution schematic figures. Follow this link and enter your details and the manuscript number to create and download figures. Upload these as the figure files for your revised submission. If you choose not to take up this offer, we require figures to be of similar quality and resolution. If you are opting out of this service to authors, state this in the Comments section on the Detailed Information page of the submission form. The link provided should only be used for the purposes of this submission. Authors will be charged for figures created on this premium BioRender account if they are not related to this manuscript submission.

This has been added.

- Please upload separate high-quality figure files via the submission form.

All figures have been uploaded separately.

- You must upload original, uncropped western blot/gel images (including controls) if they are not included in the manuscript. This is to confirm that no inappropriate, unethical or misleading image manipulation has occurred. These should be uploaded as 'Supporting information for review process only'. Please label/highlight the original gels so that we can clearly see which sections/lanes have been used in the manuscript figures. For more information, see: <https://physoc.onlinelibrary.wiley.com/hub/journal-policies#imagmanip>.

We have uploaded the original gels including the highlighted lanes that are used in the manuscript, as Supporting information for review process only.

EDITOR COMMENTS

Reviewing Editor:

Thank you for submitting your manuscript to The Journal of Physiology to be considered for inclusion in the special issue on the physiology of ageing skeletal muscle and the protective effects of exercise. Two expert reviewers evaluated your manuscript and while both were enthusiastic about the comprehensive approach and potential impact of the work, they both had several major concerns. Specifically, the introduction does not currently set the stage well for the state of the field on passive heat treatment and the different outcome variables that were studied. There is also concern that a young cohort or a time control group were not included in the experimental design and that the sample size may have resulted in being statistically underpowered for several of the outcomes.

Please also see 'Required Items' above.

We would like to thank the Reviewing Editor for the time to read and provide feedback on our manuscript. We have revised the manuscript accordingly. We have made adjustments based on the comments provided, which can be found in green in the manuscript file.

Senior Editor:

Thank you for the manuscript submission to The Journal of Physiology (TJP), which has been considered by a reviewing editor and two expert reviewers. Overall the evaluation of the manuscript was positive and although the findings reported are essentially negative the broad-array of end point measurements convey an important message. Nevertheless, a number of major and minor concerns have been raised which the authors need consider and address on a point-by-point basis when revising the manuscript if they wish the manuscript to be considered further. Thank you for considering TJP and we look forward to receiving the revised manuscript and rebuttal.

REFEREE COMMENTS

Referee #1:

The authors investigated if heating could be an intervention to improve anabolic response in older adults. They tested this possibility using a repeated measures design of anabolic response (measured by isotopic tracers), muscle capillarization and blood flow, and mitochondrial function. All of these measures have been hypothesized to be mechanisms by which heating may impact skeletal muscle, which is an important consideration in fundamental understanding of muscle physiology. The practical application of the findings is that heating did not change the muscle protein synthesis response in either basal (fasted) or in fed-state. There was an increase in capillarization, which was measured by multiple indexes, but no change in blood flow. This is an interesting finding that challenges a notion that greater capillaries drive blood flow and nutrient delivery.

The manuscript is overall very clear and methods described well. Strengths include the attention to providing a consistent meal, heating intervention and multiple complementary measurements. The discussion is overall clear integration of findings with the field. Major limitations in study design are lack of young (major) or non-heating group to account for time (minor, as it is not anticipated that there will be muscle change in parameters like synthesis rates over 8 weeks).

We would like to thank reviewer 1 for the time to read and provide feedback on our manuscript and appreciate the kind words. We have revised the manuscript accordingly. All adjustments can be found in green in the manuscript file.

Major

1) Introduction: Introduction lacks strong rationale to study heating as an intervention. As written, the intro focuses on anabolic resistance of aging, which the study design does not test because it does not include a younger group (see point 2). Currently, heating comes in the final section of the 3rd paragraph of the introduction, but is the first line of the abstract. The introduction should be revised

to better establish the state of the field of heating on blood flow, angiogenesis and mitochondrial function.

We have revised the introduction section to provide a stronger rationale for studying the impact of passive heat treatment on skeletal muscle adaptations in older individuals.

See lines 40-98: (Food intake, and dietary protein ingestion in particular, increases muscle protein synthesis rates (Rennie et al., 2002; Wolfe, 2002; Groen et al., 2015; Fuchs et al., 2019) and, as such, forms a key factor in skeletal muscle mass maintenance. The muscle protein synthetic response to food intake is blunted in older individuals (Cuthbertson et al., 2005; Katsanos et al., 2005; Wall et al., 2015), and ~~has been suggested to~~ strongly contributes to the loss of skeletal muscle mass observed with aging (Cuthbertson et al., 2005; Wilkinson et al., 2018; Fuchs et al., 2023). ~~This blunted response~~ ~~The cause of this so-called anabolic resistance~~ to protein feeding is likely multifactorial and includes a compromised postprandial increase in skeletal muscle tissue perfusion (Timmerman et al., 2010a; Timmerman et al., 2010b; Phillips et al., 2012; Phillips et al., 2015).

Adequate muscle tissue perfusion is essential in skeletal muscle mass maintenance and growth, as it allows the rapid postprandial delivery of amino acids, nutrients, and growth factors to the muscle fiber, thereby stimulating muscle protein synthesis rates. Arterial blood flow has been reported to be significantly reduced both under fasting as well as postprandial conditions in older adults (Dinenno et al., 1999; Skilton et al., 2005; Donato et al., 2006; Phillips et al., 2012). This age-related reduction in blood flow has been reported to be independent of muscle mass and may be related to chronic vasoconstriction, lower O₂ demands, and decreased endothelial wall function (Dinenno et al., 2001; Vincent et al., 2006). The delivery of nutrients to the muscle fiber is ultimately limited by the surface area of the microvascular bed (i.e. capillaries) (Pittman, 1995; Segal, 2005). We (Groen et al., 2014; Nederveen et al., 2016; Verdijk et al., 2016) as well as others (Coggan et al., 1992; Croley et al., 2005) have previously shown that muscle fiber capillarization is reduced in senescent muscle, particularly surrounding the type II muscle fibers. ~~An age-related decline in muscle fiber capillarization has been associated with elevated blood pressure (Gueugneau et al., 2016), impaired blood glucose homeostasis (Snijders et al., 2017b), decreased satellite cell function (Nederveen et al., 2018; Snijders et al., 2019; Nederveen et al., 2021), reduced physical function/exercise capacity (Nicklas et al., 2008; Prior et al., 2016; Moro et al., 2019), and lower muscle fiber growth response during prolonged exercise training (Snijders et al., 2017a; Moro et al., 2019). We hypothesised that muscle fiber capillarization and, as such muscle tissue perfusion capacity, may be responsible for a less than optimal postprandial muscle protein synthetic response in older adults. Therefore, increasing muscle fiber capillarization and, as such muscle tissue perfusion capacity, may be an effective way to augment postprandial muscle protein synthesis rates in older adults.~~

Although prolonged ~~Prolonged~~ (aerobic) exercise training is considered to be the most ~~an~~-effective intervention strategy to increase muscle fiber capillarization, and thereby improve ~~tissue vascular function/perfusion capacity in older individuals~~ (Jensen et al., 2004; Gavin et al., 2007). ~~However~~, its application is not always effective due to poor adherence or compliance to prescribed exercise programs. Therefore, we should also explore alternative therapeutic intervention strategies that may improve muscle ~~tissue~~ perfusion capacity. Passive heat treatment (PHT), which includes methods such as hot baths, steam rooms, traditional and infrared saunas, may be a promising alternative. Generally, PHT offers a range of health benefits (Patrick & Johnson, 2021), including reduced risks of cardiovascular disease and all-cause mortality (Laukkanen et al., 2015). Additionally, recent studies suggest that PHT may positively impact skeletal muscle adaptations (Kim et al., 2020a), with improvements in skeletal muscle mitochondrial adaptations (Marchant et al., 2023) and increases in muscle mass and/or strength, although the latter remains largely inconclusive (Goto et al., 2011; Kim et al., 2020b; Labidi et al., 2021). In addition, ~~has been demonstrated to increase~~ muscle fiber capillarization has been reported to increase following 6 weeks of PHT in healthy young sedentary adults, with the magnitude of change not different compared with equal time spent performing

aerobic exercise (Hesketh et al., 2019) ~~and has been reported to augment muscle tissue perfusion in vivo in calves (Saidel et al., 2001).~~ However, whether prolonged PHT can induce improvements in skeletal muscle fiber capillarization and perfusion capacity, and subsequently augment ~~The impact of PHT on muscle fiber capillarization and perfusion capacity and its potential impact on~~ basal and/or postprandial muscle protein synthesis rates *in older human adults* remains to be assessed. Therefore, ~~i~~n the present study, we recruited 14 healthy older adults in whom we investigated the impact of 8 weeks of PHT (3x per week) on muscle fiber capillarization, muscle fiber hypertrophy, and basal and postprandial muscle tissue perfusion kinetics as well as basal and postprandial muscle protein synthesis rates.)

2) Discussion: The discussion does not address limitation or context of the findings. Specifically, please address the lack of young control group and possible sex effects (such as on protein synthesis or capillarization, some reports both are higher in females).

In accordance with the reviewers' comments, we have included a limitations section in the discussion where we address the absence of a (young) control group and the lack of analysis on potential sex differences.

See lines 730-738: ("The present study is the first to provide an extensive evaluation on the skeletal muscle adaptive response to PHT in healthy older adults. Although the study included both healthy older males and females, the overall sample size was too small to assess sex-specific differences in the response to 8 weeks of PHT. Additionally, it remains to be determined whether the lack of changes in basal and/or postprandial microvascular blood flow and muscle protein synthesis rates following 8 weeks of PHT would be similarly observed in younger and/or more clinically compromised individuals. Finally, it is important to note that the study did not include a non-intervention control group. However, the primary outcomes are unlikely to change significantly over a relatively short 8-week period in the absence of a specific intervention.")

3) Results: There are variable responses in blood flow and protein synthesis. Is there correlation between blood flow or capillaries with protein synthesis, particularly the changes? It could be that heating increased protein synthesis in those that also increase capillaries/blood flow, which would align with the hypothesis. I suggest running correlation between changes in blood flow and capillaries with protein synthesis.

As suggested by the reviewer, we performed correlation analyses to investigate the relationship between changes in blood flow and fractional synthetic rate (FSR) before and after passive heat treatment. Our analysis did not reveal any significant correlation between these variables.

Furthermore, we examined the correlation between capillarization and FSR, both in the basal and postprandial states. Consistent with the previous analysis, we found no relationship between changes in capillarization and either the change in basal or postprandial FSR following PHT. However, given our small sample size, caution is warranted when interpreting such correlation analyses.

Minor

4) Suggest to temper lines about "cardiovascular health" (such as 612) since these claims seem based on a decrease in systolic BP by 3 mmHg.

Revised accordingly. Where appropriate, we have now adjusted this to “...markers of cardiovascular health...”

5) Please include number of males and females in methods and figure legends.

Revised accordingly.

6) Figures: The uses of "Ai and Aii" in figure 5 is not conventional. Please confirm with journal if this reporting aligns with journal standards.

Thank you for your suggestion. We believe that the use of "Ai and Aii" in Figure 5 aligns with the journal's standards. However, if the editorial office requests any changes, we will make the necessary adjustments.

7) Title: Suggest adding age to title, possibly as "...rates in older human"

We have the title now as follows:

“Repeated passive heat treatment increases muscle tissue capillarization, but does not affect postprandial muscle protein synthesis rates in healthy older adults”

8) Figures: Suggest adding an arrow or other indication for the meal at time 0.

We have now clearly indicated this in all figure legends.

9) Line 576: Remove paragraph break.

Revised accordingly.

10) Clarify line 378: Do you mean all samples were loaded on same gel?

The samples were loaded onto two different gels. For each participant, both the pre and post samples were always loaded on the same gel to ensure consistency. Subsequently, both gels were transferred onto the same membrane for quantification to minimize variability across samples. This has been revised within the manuscript.

See lines 386-389 (“The samples were loaded onto two different gels, with pre- and post-samples for each participant always being loaded on the same gel. Both gels were subsequently ~~All samples for each protein were~~ transferred onto the same membrane to limit variability.”).

Referee #2:

Fuchs and colleagues present a study investigating infrared sauna therapy (passive heat treatment; PHT) on muscle capillarization and muscle protein synthesis in older adults. They build upon the evidence that age-related decline in muscle perfusion is associated with impaired muscle function and anabolism and that PHT improves perfusion. Thus, it is a logical "next step" to investigate the effect of PHT on the skeletal muscle hypertrophy. Importantly, Fuchs and colleagues investigate this question in older adults who may have impaired perfusion. The investigators found that, as expected, PHT improved muscle capillarization and blood pressure and femoral vascular conductance. However, the authors indicate there were no improvements in microvascular

perfusion nor differences in the anabolic response to feeding. In all, this is a relevant investigation in the potential therapeutic potential of heat therapy.

While the study comprehensively assessed relevant physiological outcomes (blood flow, glucose/insulin concentrations, muscle FSR, and function), there is one concern regarding statistical power that should be noted.

While the field of science has a problem with biasing towards "positive" outcomes, another challenge is arriving to a false negative because inadequate power. Of course, it is difficult to anticipate the effect size of an intervention on every outcome of interest. The authors cite Timmerman et al. as evidence that perfusion to skeletal muscle can influence the anabolic response to insulin. Timmerman and colleagues infused sodium nitroprusside, a potent vasodilator in older adults along with insulin. They reported a robust increase in blood flow (Cohen's $d \sim 3.37$) and a concomitant increase in FSR (Cohen's $d \sim 2.76$). In the present study, the effect size of PHT on blood flow is moderate ($d = 0.52$) based on the reported data, which is 6 fold less than SNP. The effect size of PHT on FSR was, quite interestingly, also 0.52 (based on reported data). However, if we could assume that the effect of PHT on FSR would be similarly lower. As a result, doing a power calculation assuming an effect size of 0.50 would suggest one would need ~ 30 subjects to detect a statistically significant difference (alpha set to 0.05, powered to 0.80). Of course, this is all in hindsight. In many ways, one could consider this certainly longwinded comment as unproductive. However, this reviewer would suggest at least some commentary to be made to the fact that this is a small sample size given the intervention. Of course, the critical reader will recognize the sample size as a limitation and could likely draw their own conclusion. At the same time you are discussing the sample of the study, you could also highlight the strength of having older individuals in this study.

Despite this concern, the data presented seem to be well represented by the manuscript. Another weakness is a lack of a control group for the 8 weeks, but it is understandably difficult to include one given the time and expense.

We would like to thank reviewer 2 for the time to read and provide feedback on our manuscript and appreciate the kind words. We have revised the manuscript accordingly. All adjustments can be found in green in the manuscript file.

Specifically, we have added a section in our discussion to address the limitations raised by this reviewer and reviewer 1.

See lines 730-738: ("The present study is the first to provide an extensive evaluation on the skeletal muscle adaptive response to PHT in healthy older adults. Although the study included both healthy older males and females, the overall sample size was too small to assess sex-specific differences in the response to 8 weeks of PHT. Additionally, it remains to be determined whether the lack of changes in basal and/or postprandial microvascular blood flow and muscle protein synthesis rates following 8 weeks of PHT would be similarly observed in younger and/or more clinically compromised individuals. Finally, it is important to note that the study did not include a non-intervention control group. However, the primary outcomes are unlikely to change significantly over a relatively short 8-week period in the absence of a specific intervention.")

Minor comments:

The reviewer does have concerns with the use of utilizing paired t-tests. For example, it seems more appropriate to conduct a Two-Way ANOVA on microvascular volume, velocity, and blood flow and on muscle FSR. To compare FSR both ways (i.e. pre/post PHT and pre/post feeding), it seems a Two-

Way ANOVA would be necessary. Perhaps it would be worth checking with a statistician. However, it seems unlikely a different statistical approach would change conclusions.

The primary goal of our study was to evaluate the impact of PHT on basal and postprandial responses separately. As such we compared pre-PHT with post-PHT basal muscle protein synthesis rates. Similarly, we compared pre-PHT with post-PHT postprandial muscle protein synthesis, as outlined in the introduction. Given the nature of our research questions, we initially chose to use paired t-tests to assess the differences within subjects between pre- and post-PHT conditions.

However, we acknowledge the reviewer's point that our study includes two factors—time (basal and postprandial) and treatment (pre- and post-PHT)—which indeed justifies the consideration of a Two-Way ANOVA. To address this concern, we performed additional Two-Way ANOVA analyses on our data. However, the results from these analyses are similar those obtained from the paired t-tests, with no changes to our original conclusions.

Given that the paired t-test better aligns with the specific research questions we aimed to answer, we opted to maintain our initial statistical approach.

Here are the results from the Two-Way ANOVA analyses:

Microvascular Blood Volume:

Time effect: P=0.223

Treatment effect: P=0.035

Interaction effect: P=0.589

Microvascular Blood Velocity:

Time effect: P=0.788

Treatment effect: P=0.011

Interaction effect: P=0.946

Microvascular Blood Flow:

Time effect: P=0.560

Treatment effect: P=0.338

Interaction effect: P=0.395

FSR (Fractional Synthesis Rate):

Time effect: P=0.370

Treatment effect: P=0.126

Interaction effect: P=0.573

The authors recruited both males and females, were there any patterns of sex-specific/dimorphic responses?

Although both healthy older males and females were included in the present study, the overall sample size was too small to reliably assess any sex-specific differences in the response to 8 weeks of PHT. In the revised manuscript we included this as a limitation of the present study.

See lines 731-733: ("Although the study included both healthy older males and females, the overall sample size was too small to assess sex-specific differences in the response to 8 weeks of PHT.")

The decline in light physical activity is a surprise, is this due to the time travelling to/spent in the PHT, is the decrease in light physical activity a consequence of fatigue from the sauna?

This decline in light physical activity was indeed also a surprise to us. We cannot definitively explain the reason for this observation. The suggestions raised by the reviewer, such as the time spent traveling to or being in the PHT and potential fatigue from the sauna, are indeed possible. However, unfortunately we cannot confirm these explanations with certainty.

It is quite interesting that PHT seems to benefit those with [slightly] impaired glucose control post feeding (Figure 2B), was there any association between the relative change (%) in outcomes such as glucose and blood flow with the relative change in outcomes such as FSR, grip strength, or gait speed.

We investigated these correlations but did not find any significant relationships between the relative changes in glucose or blood flow and the outcomes such as FSR, grip strength, or gait speed.

Lines 717 - 719: the authors state that the "improved systolic blood pressure and/or postprandial glucose handling after the 8 weeks of PHT was, however, not associated with the increase in type I and type II muscle fiber capillarization." However, the authors report that capillarization in myofibers were improved (Table 5 and Figure 4). Or are the authors referring to microvascular blood flow as they conclude that microvascular blood flow does not change (Figure 5 and Lines 625-627)?

We looked at the correlations between the increase in (type I and II) muscle fiber capillarization and the improved systolic blood pressure and/or postprandial glucose handling. Here, we did not find any correlation. To make this clearer for the readers, we have changed 'associated' with 'correlated'.

See lines: 727-729 ("In the present study, the improved systolic blood pressure and/or postprandial glucose handling after the 8 weeks of PHT was, however, not ~~associated~~ ~~correlated~~ with the increase in type I and type II muscle fiber capillarization.").

Line 577 - 580: I'm confused. Do you mean: "Feeding did not elicit a change in muscle protein FSR either before ($P = 0.180$) or after ($P=0.970$) 8 weeks of PHT. In addition, 8 weeks of PHT did not change FSR in either the basal (postabsorptive) ($P=0.630$) or postprandial ($P=0.199$) state."

This is correct. In order to make this clearer for the readers we have adjusted the sentences in accordance with the suggestions made by the reviewer.

See lines: 588-593 ("~~Meal ingestion did not elicit a change in~~ ~~For~~ muscle protein FSR (~~Figure 6B~~), ~~no changes were observed following ingestion of the meal~~ either before ($P=0.180$) or after ($P=0.970$) 8 weeks of PHT (~~$P=0.180$ and $P=0.970$, respectively~~ **Figure 6B**). In addition, ~~no changes were observed in the basal (postabsorptive) and postprandial period following~~ 8 weeks of PHT ~~did not change FSR in the basal (postabsorptive; $P=0.630$) or postprandial ($P=0.630$ and $P=0.199$, respectively) state.~~").

The authors included the FSR data in the abstract, but will they also considering including it in the results of the manuscript (Lines 574 - 580)?

We included the FSR data in the abstract to provide readers with the actual numbers and their standard deviations. Since these data are already presented in the figure within the results section, we decided not to include the actual numbers in the text of the results section to avoid redundancy.

Dear Dr Snijders,

Re: JP-RP-2024-286986R1 "Repeated passive heat treatment increases muscle tissue capillarization, but does not affect postprandial muscle protein synthesis rates in humans" by Cas J Fuchs, Milan W Betz, Heather Petrick, Jil Weber, Joan Senden, Floris K Hendriks, Julia L.M. Bels, Luc JC van Loon, and Tim Snijders

Thank you for submitting your manuscript to The Journal of Physiology. It has been assessed by a Reviewing Editor and by 2 expert referees and we are pleased to tell you that it is acceptable for publication following satisfactory revision.

REVISION CHECKLIST:

We look forward to receiving your revised submission.

Yours sincerely,

Paul Greenhaff
Senior Editor
The Journal of Physiology

EDITOR COMMENTS

Reviewing Editor:

Thank you for the thorough response and diligence in revising your manuscript based on the initial feedback. Reviewer #1 is fully satisfied with the revisions, but reviewer #2 remains concerned that the study may be statistically underpowered for some of the major outcome variables. The authors are encouraged to acknowledge the possibility of a false negative for some of the outcome variables (e.g., FSR) to the limitations section in the discussion. Additionally, the full uncropped gels that were submitted do not meet the requirements of The Journal, because they do not include a molecular weight marker and loading controls. Please be sure to submit the gels as required by The Journal.

Senior Editor:

Thank you for the revised manuscript submission which has been considered by the reviewing editor and two expert reviewers that considered the original submission. All concerned believe the manuscript has been improved and to an extent that it is now provisionally acceptable for publication. However, a few points need to be addressed, including, in addition to the full uncropped gels that were submitted originally, submission of a molecular weight marker and loading controls to comply with The Journal of Physiology (TJP) publication requirements. Thank you for considering TJP. We look forward to receiving the final version of the manuscript.

REFEREE COMMENTS

Referee #1:

The authors have address my primary comments with the current version. The strengths remain their use of multiple approaches to quantify important regulatory points of muscle protein synthesis to passive heating and clear explanation of findings amidst the field.

Referee #2:

The authors have addressed my comments, thank you.

However, my concern still stands regarding the sample size for this study. The authors, in their rebuttal, report that the statistical effect of PHT on FSR was $p=0.126$, which is notable given that the sample size is only 14 and the effect size was

moderate. While I will reiterate that reporting negative results is not a problem, I'm concerned that the authors are reporting a false negative. It still seems that the authors were underpowered to detect a moderate effect size.

END OF COMMENTS

1st Confidential Review

15-Aug-2024

Response to Referees

EDITOR COMMENTS

Reviewing Editor:

Thank you for the thorough response and diligence in revising your manuscript based on the initial feedback. Reviewer #1 is fully satisfied with the revisions, but reviewer #2 remains concerned that the study may be statistically underpowered for some of the major outcome variables. The authors are encouraged to acknowledge the possibility of a false negative for some of the outcome variables (e.g., FSR) to the limitations section in the discussion. Additionally, the full uncropped gels that were submitted do not meet the requirements of The Journal, because they do not include a molecular weight marker and loading controls. Please be sure to submit the gels as required by The Journal.

We would like to thank the Reviewing Editor again for the time to read and provide feedback on our manuscript. We have made an adjustment based on the comment provided, which can be found in green in the manuscript file.

In addition, we have added a new file of the uncropped gels. The molecular weight markers have now been indicated on the full uncropped gel images. A loading control was not used because all samples were transferred and imaged on the same membrane, which limits any variability due to different transferring and/or imaging conditions. In addition, all pre- and post- samples were loaded in adjacent wells on the same gel.

Senior Editor:

Thank you for the revised manuscript submission which has been considered by the reviewing editor and two expert reviewers that considered the original submission. All concerned believe the manuscript has been improved and to an extent that it is now provisionally acceptable for publication. However, a few points need to be addressed, including, in addition to the full uncropped gels that were submitted originally, submission of a molecular weight marker and loading controls to comply with The Journal of Physiology (TJP) publication requirements. Thank you for considering TJP. We look forward to receiving the final version of the manuscript.

REFEREE COMMENTS

Referee #2:

The authors have addressed my comments, thank you.

However, my concern still stands regarding the sample size for this study. The authors, in their rebuttal, report that the statistical effect of PHT on FSR was $p=0.126$, which is notable given that the sample size is only 14 and the effect size was moderate. While I will reiterate that reporting negative results is not a problem, I'm concerned that the authors are reporting a false negative. It still seems that the authors were underpowered to detect a moderate effect size.

We would like to thank Reviewer 2 again for their time and thoughtful comment. We acknowledge the concern raised by this reviewer, and we have revised the manuscript accordingly to address this point.

See lines: 718-721: (“Additionally, it could be speculated that the sample size was insufficient to detect a statistically significant change in postprandial muscle protein synthesis rates following PHT, as a moderate effect size ($d=0.53$) was observed for this specific outcome parameter.”)

Dear Dr Snijders,

Re: JP-RP-2024-286986R2 "Repeated passive heat treatment increases muscle tissue capillarization, but does not affect postprandial muscle protein synthesis rates in humans" by Cas J Fuchs, Milan W Betz, Heather Petrick, Jil Weber, Joan Senden, Floris K Hendriks, Julia L.M. Bels, Luc JC van Loon, and Tim Snijders

We are pleased to tell you that your paper has been accepted for publication in The Journal of Physiology.

Authors should note that it is too late at this point to offer corrections prior to proofing. Major corrections at proof stage, such as changes to figures, will be referred to the Editors for approval before they can be incorporated. Only minor changes, such as to style and consistency, should be made at proof stage. Changes that need to be made after proof stage will usually require a formal correction notice.

If you would like to receive our 'Research Roundup', a monthly newsletter highlighting the cutting-edge research published in The Physiological Society's family of journals (The Journal of Physiology, Experimental Physiology and Physiological Reports), please click this link, fill in your name and email address and select 'Research Roundup': <https://www.physoc.org/journals-and-media/membernews/>.

Yours sincerely,

Paul Greenhaff
Senior Editor
The Journal of Physiology

P.S. - You can help your research get the attention it deserves! Check out Wiley's free Promotion Guide for best-practice recommendations for promoting your work at www.wileyauthors.com/eeo/guide. You can learn more about Wiley Editing Services which offers professional video, design, and writing services to create shareable video abstracts, infographics, conference posters, lay summaries, and research news stories for your research at www.wileyauthors.com/eeo/promotion.

IMPORTANT NOTICE ABOUT OPEN ACCESS: To assist authors whose funding agencies mandate public access to published research findings sooner than 12 months after publication, The Journal of Physiology allows authors to pay an Open Access (OA) fee to have their papers made freely available immediately on publication.

You can check if your funder or institution has a Wiley Open Access Account here: <https://authorservices.wiley.com/author-resources/Journal-Authors/licensing-and-open-access/open-access/author-compliance-tool.html>.

EDITOR COMMENTS

Reviewing Editor:

Thank you for submitting the remaining revisions to your manuscript. The revisions are all satisfactory, and I would like to congratulate the authors on the completion of an excellent study.

Senior Editor:

Thank you for making the requested minor revisions and supplying the uncropped gels for consideration (which is standard practice for The Journal of Physiology). The manuscript is now deemed to be acceptable for publication. Thank you for considering The Journal of Physiology to publish your research.

2nd Confidential Review

11-Sep-2024